# Verifying Properties of Binary Neural Networks Using Sparse Polynomial Optimization

**Jianting Yang** [*]
Academy of Mathematics and Systems Science
Chinese Academy of Sciences
yangjianting@amss.ac.cn

**Srećko Đurašinović**
CNRS@CREATE Singapore
CCDS, NTU Singapore
srecko001@e.ntu.edu.sg

**Jean-Bernard Lasserre**
LAAS-CNRS
Toulouse School of Economics
lasserre@laas.fr

**Victor Magron**
LAAS-CNRS
Université de Toulouse
vmagron@laas.fr

**Jun Zhao**
CCDS, NTU Singapore
junzhao@ntu.edu.sg

## Abstract

This paper explores methods for verifying the properties of Binary Neural Networks (BNNs), focusing on robustness against adversarial attacks. Despite their lower computational and memory needs, BNNs, like their full-precision counterparts, are also sensitive to input perturbations. Established methods for solving this problem are predominantly based on Satisfiability Modulo Theories and Mixed-Integer Linear Programming techniques, which often face scalability issues. We introduce an alternative approach using Semidefinite Programming relaxations derived from sparse Polynomial Optimization. Our approach, compatible with continuous input space, not only mitigates numerical issues associated with floating-point calculations but also enhances verification scalability through the strategic use of tighter first-order semidefinite relaxations. We demonstrate the effectiveness of our method in verifying robustness against both $\|.\|_\infty$ and $\|.\|_2$-based adversarial attacks.

## 1 Introduction

In the evolving landscape of machine learning, Binary Neural Networks (BNNs) have emerged as an intriguing class of neural networks since their introduction in 2015 (Courbariaux et al., 2015; Hubara et al., 2016). The architectural simplicity and inherent advantages of BNNs, such as reduced memory requirements and lower computational time, have garnered significant attention. As these networks have matured, they have been integrated into a wide range of machine learning applications (Rastegari et al., 2016; Sun et al., 2018; Xiang et al., 2017; Vorabbi et al., 2024). For instance, BNNs are widely used in edge devices for tasks such as object detection, image recognition, and decision-making in resource-constrained environments, like autonomous delivery drones.

Despite their great approximation capacities, general deep neural networks are sensitive to input perturbations (Szegedy et al., 2014; Antun et al., 2021). BNNs are not any different in this regard. Indeed, quantized or binarized networks do not necessarily preserve the properties satisfied by their real-precision counterparts (Galloway et al., 2018; Giacobbe et al., 2020). BNN-controlled autonomous delivery drones, for example, might encounter adversarial environmental conditions, such as lighting variations, or weather disturbances. Failing to robustly classify objects under these altered environmental conditions could lead to a collision, endangering people or property. That is why verifying their robustness is one of the crucial aspects of their design and deployment. A classifying network is said to be robust to adversarial attacks if small input perturbations do not cause any misclassifications. Formally speaking, the network verification consists of finding a point $x$ satisfying $P(x) \wedge Q(\mathbf{BNN}(x))$, where $P$ is a pre-condition on $x$, e.g., stating a valid input perturbation, and $Q$ is a post-condition on $y = \mathbf{BNN}(x)$, e.g., stating an (undesirable) alteration of the highest output score. If both properties hold for some $x$, the network is not robust.

---

[*]Jianting Yang was mainly affiliated with CNRS@CREATE Singapore during the completion of this paper.

Compared to much research on the robustness verification of full-precision networks, verification of BNNs is a question yet to be addressed with more attention. Moreover, many formal verification frameworks do not exploit the bit-precise semantics of BNNs (Giacobbe et al., 2020), and in parallel, those BNN-specific approaches for robustness verification often suffer from limited scalability (Lazarus & Kochenderfer, 2022; Narodytska et al., 2020). Two related methodological axes can be identified. Firstly, methods based on Satisfiability Modulo Theories (SAT/SMT), as in Amir et al. (2021); Jia & Rinard (2020a); Narodytska et al. (2018; 2020), encode the BNN verification problem into Boolean formula satisfiability, and leverage modern off-the-shelf solvers to prove robustness or find counterexamples. Secondly, Khalil et al. (2019); Lazarus & Kochenderfer (2022) cast robustness verification as a Mixed Integer Linear Programming (MILP) optimization problem. Both approaches belong to the group of exact verification methods, i.e., they are sound and complete. In this paper, instead of solving the BNN verification problem exactly, we rather examine it through the lens of (Sparse) Polynomial Optimization Problems (POP) (Lasserre, 2015; Magron & Wang, 2023), that can be approximated with hierarchies of relaxations based on Semidefinite Programming (SDP). Recent frameworks (Chen et al., 2020; 2021; Latorre et al., 2020; Newton & Papachristodoulou, 2023) have demonstrated that sparse variants of SDP hierarchies could certify the robustness of full-precision ReLU neural networks, by efficiently providing accurate bounds for the associated optimization problems.

## 1.1 Contribution

- We exploit the semi-algebraic nature of the sign activation function to encode the BNN verification problem as a POP. We then solve the resulting first-order SDP relaxation of this POP to obtain lower bounds that can certify robustness. In addition, our method overcomes floating-point-related numerical issues that typically compromise the branch and bound process of MILP solvers. To the best of our knowledge, this is the first SDP-based method for BNN verification.

- From the theoretical point of view, we prove that adding tautologies (redundant constraints) to the initial POP encoding leads to first-order SDP relaxations with highly improved accuracy. Knowing that higher-order SDP relaxations quickly become intractable for high-dimensional problems, designing tighter first-order SDP relaxations that exploit the structure of the network is crucial for enhancing the scalability of the method. We show that our bounds can be up to $55\%$ more accurate than those derived from the linear relaxations typically used in traditional MILP algorithms.

- We demonstrate the effectiveness of our method, compatible with continuous input space, in verifying robustness against both $\|.\|_\infty$ and $\|.\|_2$-based adversarial attacks, the latter being much less studied in the BNN verification literature. Our experimental results indicate that, for $\|.\|_\infty$ and $\|.\|_2$ robustness verification problems, our algorithm can provide an average speedup of $4.5$ and $11.4$ times, respectively. For some severe attacks, the speedup exceeds a factor of 50.

## 1.2 Related works

**General verification methods:** Neural network verification has been extensively studied in recent years. Optimization solvers based on works of Xu et al. (2021); Wang et al. (2021b) integrate bound propagation with efficient gradient computations, achieving state-of-the-art performance on multiple benchmarks. Reachability-based verification using abstract domain representations has been explored in Lopez et al. (2023), while an enhanced abstract interpretation approach has been recently implemented in PyRAT (Lemesle et al., 2024). Verification via MILP formulations has been investigated in Katz et al. (2019). Sampling-based approach has been introduced in Li et al. (2022). An overview of established verification techniques is available in Liu et al. (2021).

**SDP-based verification methods:** Certifying adversarial robustness using SDP relaxations has been first proposed in Raghunathan et al. (2018). Tightening of the SDP bounds via linear reformulations and quadratic constraints has been proposed in Fazlyab et al. (2020); Batten et al. (2021); Lan et al. (2022). Verification can also be tackled by computing upper bounds of Lipschitz constants (Fazlyab et al., 2019; Latorre et al., 2020; Chen et al., 2020). Chordal and correlative sparsity can be exploited (Anton et al., 2024; Newton & Papachristodoulou, 2023; Chen et al., 2021) to design more efficient SDP relaxations. The first-order dual SDP approach developed in Dathathri et al. (2020) enables efficient verification of high-dimensional models. Note that these approaches have been mainly designed to verify standard full-precision ReLU networks.

**BNN verification:** Binarized weights and activation functions allow for the BNN verification problem to be exactly encoded into a Boolean expression. However, solving the resulting encoding via SAT solvers is usually quite computationally expensive as the number of involved variables grows very fast with the network size (Narodytska et al., 2018; Narodytska, 2018). Consequently, only small and medium-sized networks can be handled within this framework. In order to improve the scalability, different architectural and training choices have been proposed: inducing sparse weight matrices, achieving neuron stabilization via input bound propagation, direct search instead of unguided jumps for clause-conflict resolution (Narodytska et al., 2020), enforcing sparse patterns to increase the number of shared computations, neuron-factoring (Narodytska et al., 2020; Cheng et al., 2018). The SAT solver proposed in Jia & Rinard (2020a;b) is tailored for BNN verification. By efficiently handling reified cardinality constraints and exploiting balanced weight sparsification, this solver could achieve significantly faster verification on large networks. The SMT-based framework from Amir et al. (2021) extends the well-known Reluplex (Katz et al., 2017) framework to support binary activation functions. Notice that these works are either incompatible with continuous input data or are restricted to $\|.\|_\infty$ perturbations. In contrast, our approach does not require an extra input quantization step and is compatible with more general perturbation regions, including the one defined by the $\|.\|_2$ norm.

The sole nature of BNNs makes them convenient for being directly represented via linear inequalities and binary variables, enabling the verification problem to be cast as a MILP. These properties are exploited in Khalil et al. (2019), where *big-M constraints* model the neuron activations, and a heuristic named *IProp* decomposes the original problem into smaller MILP problems. However, the big-$M$ approach is also well-known to be sensitive to the $M$ parameter (Gurobi Optimization, LLC). Improved bounds for each neuron have been proposed by Han & Goméz (2021) that lead to tighter relaxations, but only on medium-sized networks. This suggests that BNNs might also suffer from a convex relaxation barrier for tight verification (Salman et al., 2019). Another MILP-based approach has been discussed in Lazarus & Kochenderfer (2022); Lazarus et al. (May, 2022), where both input and output domain of a property to be verified are restricted to polytopes. The set-reachability method from Ivashchenko et al. (2023) extends the *star* set and *Image star* approaches from Bak & Duggirala (2017); Tran et al. (2019; 2020). Unlike most other methods, it allows the input space to be continuous. However, our experiments highlight the intrinsic weaknesses of LP relaxations for BNN verification, negatively influencing the bounding step of MILP solving. We argue that SDP bounds would be able to provide significant speedups to these exact solvers, especially for larger networks and more severe attacks. Somewhat related works concern *quantitative* BNN verification, where one tries to estimate how often a given network satisfies or violates some property. In Baluta et al. (2019); Narodytska et al. (2019), quantitative robustness verification is reduced into a model counting problem over a Conjunctive Normal Form (CNF) expression. SAT-based approaches are derived from either (Ordered) Binary or Sentential Decision Diagrams in Shi et al. (2020); Shih et al. (2019); Zhang et al. (2021).

## 1.3 NOTATIONS AND PRELIMINARIES

We use Roman letters to denote scalars, and boldfaced letters to represent vectors and matrices. If $\boldsymbol{A}$ is a matrix, then $\boldsymbol{A}_{(k,:)}$ denotes its $k$-th row vector, and $\|\boldsymbol{A}\|_F$ denotes its Frobenius norm. The entry $j$ of a vector $\boldsymbol{x}$ (or $\boldsymbol{x}_i$) is denoted by $\boldsymbol{x}_j$ (or $\boldsymbol{x}_{i,j}$). The Hadamard product is denoted by $\odot$, i.e., $(\boldsymbol{A} \odot \boldsymbol{B})_{(i,j)} = \boldsymbol{A}_{(i,j)} \boldsymbol{B}_{(i,j)}$. For any $p \times q$ matrix $\boldsymbol{A}$, $\mathrm{nv}(\boldsymbol{A})$ is a $p$-dimensional column vector whose $k$-th coordinate is given by $\|\boldsymbol{A}_{(k,:)}\|_1$.

The ring of $n$-variate real polynomials (resp. of at most degree $d$) is denoted by $\mathbb{R}[\boldsymbol{x}]$ (resp. $\mathbb{R}[\boldsymbol{x}]_d$). Let $\Sigma[\boldsymbol{x}]$ be the cone of multivariate Sum Of Squares (SOS) polynomials and $\Sigma[\boldsymbol{x}]_d := \Sigma[\boldsymbol{x}] \cap \mathbb{R}[\boldsymbol{x}]_{2d}$. By $\langle \boldsymbol{x}, \boldsymbol{y} \rangle = \sum_{i=1}^n x_i y_i$ we denote the standard inner product of vectors $\boldsymbol{x}, \boldsymbol{y} \in \mathbb{R}^n$. If $k_1, k_2 \in \mathbb{N}$ with $k_1 \leq k_2$, then $[\![k_1, k_2]\!] := \{k_1, \ldots, k_2\}$. Let $\mathbb{B}_{\|.\|}(\bar{\boldsymbol{x}}, \varepsilon)$ be the $\|.\|$-ball of radius $\varepsilon$ centered at $\bar{\boldsymbol{x}}$. We denote by $\mathbb{S}_+^n$ the set of symmetric positive semidefinite matrices of size $n$. We recall that $f \in \mathbb{R}[\boldsymbol{x}]_{2d}$ is SOS if and only if a positive semidefinite matrix $\boldsymbol{G}$ satisfies $f = \boldsymbol{v}_d^\mathsf{T} \boldsymbol{G} \boldsymbol{v}_d$, where $\boldsymbol{v}_d := (1, x_1, \ldots, x_n, \ldots, x_n^d)^\mathsf{T}$ is the vector of monomials of degree at most $d$ with size $s(d) := \binom{n+d}{d}$.

**Definition 1.1.** *Let $\boldsymbol{g} := (g_j)_{j \in [\![1,m]\!]}$ and $\boldsymbol{h} := (h_k)_{k \in [\![1,l]\!]}$ denote families of polynomial functions. For all $j \in [\![1, m]\!]$, $k \in [\![1, l]\!]$, define $d_j := \lceil \deg(g_j)/2 \rceil$ and $\bar{d}_k := \deg(h_k)$. Then, the $d$-truncated*

*quadratic module $\mathcal{Q}_d(\boldsymbol{g})$ generated by $\boldsymbol{g}$, and the $d$-truncated ideal $\mathcal{I}_d(\boldsymbol{h})$ generated by $\boldsymbol{h}$ are*

$$\mathcal{Q}_d(\boldsymbol{g}) := \left\{ \sigma_0 + \sum_{j=1}^{m} \sigma_j g_j \mid \sigma_0 \in \Sigma[\boldsymbol{x}]_d, \sigma_j \in \Sigma[\boldsymbol{x}]_{d-d_j}, j \in [\![1, m]\!] \right\}, \tag{1}$$

$$\mathcal{I}_d(\boldsymbol{h}) := \left\{ \sum_{k=1}^{l} \psi_k h_k, \psi_k \in \mathbb{R}[\boldsymbol{x}]_{2d-\bar{d}_k}, k \in [\![1, l]\!] \right\}. \tag{2}$$

## 2 MAIN INGREDIENTS

### 2.1 BINARY NEURAL NETWORKS

Let $L \geq 1$ be the number of hidden layers of a classifying BNN, with layer widths being given by $\boldsymbol{n} = (n_0, n_1, \ldots, n_L, n_{L+1})^{\mathsf{T}} \in \mathbb{N}^{L+2}$, where $n_0$ and $n_{L+1}$ are input and output dimensions. A feed-forward BNN is a mapping from the input region $\mathcal{R}_{n_0} \subset \mathbb{R}^{n_0}$ to the output set $[\![1, n_{L+1}]\!]$ realized via successive compositions of several internal blocks $(\mathbf{B}_i)_{i=1,\ldots,L}$ and an output block $\mathbf{B}_o$:

$$\begin{aligned} \text{BNN} : \mathcal{R}_{n_0} &\to [\![1, n_{L+1}]\!] \\ \boldsymbol{x}_0 &\mapsto \text{BNN}(\boldsymbol{x}_0) := \text{argmax} \left( \mathbf{B}_o(\mathbf{B}_L(\ldots(\mathbf{B}_1(\boldsymbol{x}_0)))) \right). \end{aligned} \tag{3}$$

For any $i \in [\![1, L]\!]$, the internal block $\mathbf{B}_i$ implements successively three different operations: affine transformation, batch normalization[1] and point-wise binarization, so that its output vector, denoted by $\boldsymbol{x}_i$, belongs to $\{-1, 1\}^{n_i}$. These operations are described by a set of trainable parameters:

$$\left( \boldsymbol{W}^{[i+1]}, \boldsymbol{b}^{[i+1]} \right)_{i \in [\![0, L]\!]} \in \{-1, 0, 1\}^{n_{i+1} \times n_i} \times \mathbb{R}^{n_{i+1}}, \tag{4}$$

$$\left( \boldsymbol{\gamma}^{[i]}, \boldsymbol{\beta}^{[i]}, \boldsymbol{\mu}^{[i]}, \boldsymbol{\sigma}^{2,[i]} \right)_{i \in [\![1, L]\!]} \in (\mathbb{R}^{n_i})^4 . \tag{5}$$

Consequently, the output of a neuron $j \in [\![1, n_i]\!]$ from the hidden layer $i \in [\![1, L]\!]$ is given by $\boldsymbol{x}_{i,j} = \text{sign} \left( \boldsymbol{\gamma}_j^i \frac{\left( \langle \boldsymbol{W}_{(j,:)}^{[i]}, \boldsymbol{x}_{i-1} \rangle + \boldsymbol{b}_j^{[i]} - \boldsymbol{\mu}_j^{[i]} \right)}{\sqrt{\boldsymbol{\sigma}_j^{2,[i]} + \varepsilon}} - \boldsymbol{\beta}_j^i \right)$, with small enough $\varepsilon > 0$. The output block $\mathbf{B}_o$ applies a softmax transformation to the affinely-transformed outputs of the last hidden layer, i.e., for each $j \in [\![1, n_{L+1}]\!]$, $\boldsymbol{x}_{L+1,j} = \frac{\exp(z_j)}{\sum_{k=1}^{n_{L+1}} \exp(z_k)}$, where $z_j = \boldsymbol{W}_{(j,:)}^{[L+1]} \boldsymbol{x}_L + \boldsymbol{b}_j^{[L+1]}$.

In BNNs, replacing vector-matrix multiplications with simpler 1-bit XNOR-count operations comes at a cost: the sign function prevents effective (adversarial) training due to the lack of proper gradient information. However, a promising framework for training BNNs via automatic differentiation was recently introduced in Aspman et al. (A2024). For a comprehensive analysis and the latest advancements in training (robust) BNNs, see Qin et al. (2020); Yuan & Agaian (2021).

### 2.2 PROBLEM FORMULATION

Casting BNN verification as an optimization problem requires using an appropriate representation of the non-linear $\text{sign}(\cdot)$ activation function. For any $(a, b) \in \mathbb{R}^2$, we have:

$$a = \text{sign}(b) \implies a^2 - 1 = 0, \text{ and } ab \geq 0. \tag{6}$$

This motivates the introduction of a sequence of vector-valued functions $(\boldsymbol{h}_i, \boldsymbol{g}_i)_{i \in [\![1, L]\!]}$ such that

$$\boldsymbol{x}_i := \text{sign} \left( \boldsymbol{W}^{[i]} \boldsymbol{x}_{i-1} + \boldsymbol{b}^{[i]} \right) \implies \begin{cases} \boldsymbol{h}_i(\boldsymbol{x}_i) := \boldsymbol{x}_i \odot \boldsymbol{x}_i - \mathbf{1} = \mathbf{0}, & \text{(7a)} \\ \boldsymbol{g}_i(\boldsymbol{x}_i, \boldsymbol{x}_{i-1}) := \boldsymbol{x}_i \odot (\boldsymbol{W}^{[i]} \boldsymbol{x}_{i-1} + \boldsymbol{b}^{[i]}) \geq \mathbf{0}, & \text{(7b)} \end{cases}$$

where $\boldsymbol{b}^{[i]} \in \mathbb{R}^{n_i}$ satisfies $\left| \boldsymbol{b}_k^{[i]} \right| < \text{nv} \left( \boldsymbol{W}^{[i]} \right)_k$ for each $k \in [\![1, n_i]\!]$. This assumption eliminates the case in which some neurons are either always or never activated, since such neurons have no impact

---

[1]Since batch normalization can be understood as another affine transformation, it can be omitted throughout the technical modeling part, without loss of generality.

on the verification process. Furthermore, we suppose that the input perturbation region $\boldsymbol{B} \subseteq \mathbb{R}^{n_0}$ can be encoded via positivity conditions on (at most quadratic) polynomials $\boldsymbol{x}_0 \mapsto \boldsymbol{g}_{\boldsymbol{B}}(\boldsymbol{x}_0)$. For example, $\boldsymbol{g}_{\boldsymbol{B}}(\boldsymbol{x}_0) = (\boldsymbol{\varepsilon} + \bar{\boldsymbol{x}} - \boldsymbol{x}_0) \odot (\boldsymbol{\varepsilon} - \bar{\boldsymbol{x}} + \boldsymbol{x}_0)$ corresponds to $\boldsymbol{B} = \mathbb{B}_{||\cdot||_\infty}(\bar{\boldsymbol{x}}, \varepsilon)$, where $\bar{\boldsymbol{x}} \in \mathcal{R}_{n_0}$. Finally, the *standard* form BNN verification problems studied here are:

$$
\tau := \begin{cases} \min_{\boldsymbol{x}_0, \boldsymbol{x}_1, \ldots, \boldsymbol{x}_L} f(\boldsymbol{x}_0, \boldsymbol{x}_1, \ldots, \boldsymbol{x}_L) & \text{(8a)} \\ \text{s.t.} \quad \boldsymbol{h}_i(\boldsymbol{x}_i) = \boldsymbol{0}, \ i \in [\![1, L]\!], & \text{(8b)} \\ \quad\quad \boldsymbol{g}_i(\boldsymbol{x}_i, \boldsymbol{x}_{i-1}) \geq \boldsymbol{0}, \ i \in [\![1, L]\!], & \text{(8c)} \\ \quad\quad \boldsymbol{g}_{\boldsymbol{B}}(\boldsymbol{x}_0) \geq \boldsymbol{0}, & \text{(8d)} \end{cases}
$$

where $f$ is either a linear or a quadratic function.

**Remark 2.1** (Adversarial attacks). *Given some point $\bar{\boldsymbol{x}} \in \mathcal{R}_{n_0}$ whose true label is $\bar{y} \in [\![1, n_{L+1}]\!]$, we define a $k$-targeted attack to be any allowed perturbation $\boldsymbol{x}_0$ satisfying (8b)-(8d) for which the output of the network is $k \neq \bar{y}$. If the network is robust, then*

$$
f_k^{\mathrm{adv}}(\boldsymbol{x}_0, \boldsymbol{x}_1, \ldots, \boldsymbol{x}_L) := \left\langle \boldsymbol{W}_{(\bar{y},:)}^{[L+1]} - \boldsymbol{W}_{(k,:)}^{[L+1]}, \boldsymbol{x}_L \right\rangle + b_{\bar{y}}^{[L+1]} - b_k^{[L+1]} \tag{9}
$$

*is always positive. Notice that $f_k^{\mathrm{adv}}$ is an affine mapping of the neurons from the last hidden layer. Our framework is suitable for verifying properties of BNNs other than adversarial robustness (e.g. certain properties describing the ACAS-Xu controller from Katz et al. (2017), or energy conservation in dynamical systems (Qin et al., 2019)). Verification against non-targeted attacks could be achieved via a simple objective function modification.*

### 2.3 SPARSE POLYNOMIAL OPTIMIZATION

Notice that (8) is an instance of Quadratically Constrained Quadratic Programming (QCQP), involving $n := \sum_{i=0}^{L} n_i$ decision variables and $m := 2 \sum_{i=1}^{L} n_i + n_{\boldsymbol{B}}$ constraints, where $n_{\boldsymbol{B}}$ is the number of polynomials (at most quadratic) needed to represent the input region $\boldsymbol{B}$. As such, problem (8) is a special case of Polynomial Optimization since one minimizes a polynomial $f$ over a feasible set $S$ defined with finitely many polynomial (in)equality constraints. Here $S := \{\boldsymbol{x} \mid \boldsymbol{g}_i(\boldsymbol{x}) \geq \boldsymbol{0}, \boldsymbol{h}_i(\boldsymbol{x}) = \boldsymbol{0}, \forall i \in [\![1, L]\!], \boldsymbol{g}_{\boldsymbol{B}}(\boldsymbol{x}_0) \geq \boldsymbol{0}\}$. Note that an equivalent characterization of the global infimum of $f$ on $S$ is $\tau = \min\{f(\boldsymbol{x}) \mid \boldsymbol{x} \in S\} = \max\{\lambda \in \mathbb{R} \mid f - \lambda \geq 0 \text{ on } S\}$. This requires one to efficiently handle the set of polynomials that are nonnegative on $S$, which is known to be intractable. However, in practice, we can rely on its *tractable* inner approximations based on weighted combinations of elements in $\boldsymbol{g} := \{(\boldsymbol{g}_i)_{i \in [\![1,L]\!]}, \boldsymbol{g}_{\boldsymbol{B}}\}$, the weights being SOS polynomials.

By (Lasserre, 2001, Theorem 4.2), $\tau^d := \sup\{\lambda \in \mathbb{R} \mid f - \lambda - \sigma \in \mathcal{I}_d(\boldsymbol{h}), \sigma \in \mathcal{Q}_d(\boldsymbol{g})\}$ defines a hierarchy of *dense* SDP relaxations whose size increases with the relaxation order $d$, and such that $\tau^d \uparrow \tau$ as $d \to +\infty$. Moreover, the convergence towards $\tau$ is generically finite, which means that $\tau^d = \tau$ for some $d \in \mathbb{N}$ (Nie, 2014, Theorem 1.1).

**Example 2.1.** *Consider a BNN with $L = 2$, $(n_0, n_1, n_2, n_3) = (3, 2, 2, 2)$, with trainable parameters given by $\left(\boldsymbol{W}^{[1]}, \boldsymbol{W}^{[2]}, \boldsymbol{W}^{[3]}\right) = \left(\begin{pmatrix} -1 & 1 & 1 \\ -1 & -1 & 1 \end{pmatrix}, \begin{pmatrix} -1 & -1 \\ -1 & 1 \end{pmatrix}, \begin{pmatrix} -1 & 1 \\ -1 & -1 \end{pmatrix}\right)$, and $\left(\boldsymbol{b}^{[1]}, \boldsymbol{b}^{[2]}, \boldsymbol{b}^{[3]}\right) = \left(\begin{pmatrix} 1.5 \\ 2 \end{pmatrix}, \begin{pmatrix} 1 \\ -0.5 \end{pmatrix}, \begin{pmatrix} -2 \\ -1 \end{pmatrix}\right)$. Suppose $g_{\boldsymbol{B}}(\boldsymbol{x}_0) = 0.2^2 - (\boldsymbol{x}_0 - \bar{\boldsymbol{x}})^\mathsf{T}(\boldsymbol{x}_0 - \bar{\boldsymbol{x}})$, with $\bar{\boldsymbol{x}} = (0, 0.5, 0)^\mathsf{T}$. The network assigns the label $\bar{y} = 2$ since $\boldsymbol{x}_{3,2} = 1 > \boldsymbol{x}_{3,1} = -2$. By Remark 2.1, the affine objective function to minimize becomes $\boldsymbol{x} \mapsto f_1^{\mathrm{adv}}(\boldsymbol{x}) = -(-\boldsymbol{x}_{2,1} - \boldsymbol{x}_{2,2} + \boldsymbol{x}_{2,1} - \boldsymbol{x}_{2,2} - 1 + 2) = 2\boldsymbol{x}_{2,2} - 1$. The corresponding dense SDP relaxation of order $d \geq 1$ becomes*

$$
\tau^d = \begin{cases} \sup_{\lambda, \{\boldsymbol{G}_{i,j}\}_{i,j=1}^2, \boldsymbol{G}_0, \boldsymbol{G}_{\boldsymbol{B}}} \lambda \\ \text{s.t.} \quad f_1^{\mathrm{adv}} - \lambda - \sigma \in \mathcal{I}_d(\{\boldsymbol{h}_1, \boldsymbol{h}_2\}), & \text{(10a)} \\ \quad \sigma = \boldsymbol{v}_d^\mathsf{T} \boldsymbol{G}_0 \boldsymbol{v}_d + \sum_{i,j=1}^2 \boldsymbol{v}_{d-1}^\mathsf{T} \boldsymbol{G}_{i,j} \boldsymbol{v}_{d-1} g_i(\cdot)_j + \boldsymbol{v}_{d-1}^\mathsf{T} \boldsymbol{G}_{\boldsymbol{B}} \boldsymbol{v}_{d-1} g_{\boldsymbol{B}}, & \text{(10b)} \\ \quad \boldsymbol{G}_{\boldsymbol{B}}, \boldsymbol{G}_{i,j} \in \mathbb{S}_+^{s(d-1)}, i, j \in [\![1, 2]\!], \boldsymbol{G}_0 \in \mathbb{S}_+^{s(d)}. & \text{(10c)} \end{cases}
$$

*In addition to SDP conditions from (10c), checking for membership in $\mathcal{I}_d(\{\boldsymbol{h}_1, \boldsymbol{h}_2\})$ in the line (10a) boils down to imposing linear conditions on the coefficients of the involved polynomials. Those can be obtained after applying the substitution rules $\boldsymbol{h}_i(\boldsymbol{x}_i) = 0 \iff \boldsymbol{x}_{i,j}^2 = 1$ for $i, j \in [\![1, 2]\!]$.*

For a fixed relaxation order $d$, dense SDP relaxations involve $\mathcal{O}(n^{2d})$ equality constraints, which prevents them from being applied to large-scale problems. However, many POP problems, including BNN robustness verification, exhibit important structural sparsity properties, which enables one to build significantly more computationally efficient SDP relaxations (Waki et al., 2006; Wang et al., 2021a). For instance, let us suppose that $[\![1, n]\!] =: I_0 = \cup_{k=1}^p I_k$ with $I_k$ not necessarily disjoint. To every subset $I_k$ we can associate a *subset of decision variables* $\boldsymbol{x}_{I_k} := \{x_i, i \in I_k\}$. An instance of the BNN robustness verification problem of the form (8) exhibits *correlative sparsity* since

- There exist $(f_k)_{k \in [\![1, p]\!]}$ such that $f = \sum_{k=1}^p f_k$, with $f_k \in \mathbb{R}[\boldsymbol{x}_{I_k}]$,
- The polynomials $\boldsymbol{g}$ can be split into disjoints sets $J_k$, such that $\boldsymbol{g}_i(\cdot)_j \in J_k$ if and only if $\boldsymbol{g}_i(\cdot)_j \in \mathbb{R}[\boldsymbol{x}_{I_k}]$. Moreover, $\boldsymbol{g}_{\boldsymbol{B}} \in J_k$ for $k \in [\![1, p]\!]$. Since $\boldsymbol{h}_i(\cdot)_j$ only depends on $\boldsymbol{x}_{i,j}$, the overall sparsity structure is induced by inequality constraints that mimic the cascading BNN structure.

As in the dense case, a hierarchy of correlatively sparse SDP relaxations is given by $\tau_{\text{cs}}^d := \sup\{\lambda \in \mathbb{R} \mid f - \lambda - \sum_{k=1}^p \sigma_k \in \mathcal{I}_d(\boldsymbol{h}), \sigma_k \in \mathcal{Q}_d(\{\boldsymbol{g}_i(\cdot)_j \in J_k\})\}$. Under additional ball constraints (Magron & Wang, 2023, Assumption 3.1), one still has $\tau_{\text{cs}}^d \uparrow \tau$ as $d \to +\infty$. If $\rho := \max_k |I_k|$, then these sparse relaxations involve $\mathcal{O}(p\rho^{2d})$ equality constraints, yielding a significant improvement when $\rho \ll n$. Apart from this computational gain, we will benefit from the fact that the first-order sparse relaxation is not conservative with respect to the dense one, meaning that $\tau_{\text{cs}}^1 = \tau^1$ (Vandenberghe et al., 2015, Theorem 9.2).

## 3 COMPARISON OF LINEAR PROGRAMMING (LP) AND SDP BOUNDS

Here, we assume that $f$ is linear, e.g., the function $f_k^{\text{adv}}$ defined in (9). The goal of this section is to compare LP and SDP relaxations of the QCQP encoding (8), when the perturbation region is described by $\boldsymbol{B} = \mathbb{B}_{||.||_\infty}(\bar{\boldsymbol{x}}, \varepsilon) = \{\boldsymbol{x} \mid \bar{\boldsymbol{x}} - \boldsymbol{\varepsilon} \leq \boldsymbol{x} \leq \boldsymbol{\varepsilon} + \bar{\boldsymbol{x}}\}$. Let $n \geq 2$, $\boldsymbol{w} \in \{-1, 1\}^n$, and $b \in \mathbb{R}$ satisfying $|b| < n$. As pointed out by Amir et al. (2021), an equivalent linear approximation of the set $\{(\boldsymbol{x}, y) \in [-1, 1]^n \times \{-1, 1\}, y = \text{sign}(\langle \boldsymbol{w}, \boldsymbol{x} \rangle + b)\}$ is given by

$$\left\{(\boldsymbol{x}, y) \in [-1, 1]^n \times \{-1, 1\}, y \geq \frac{2(\langle \boldsymbol{x}, \boldsymbol{w} \rangle + b)}{n + b} - 1, \ y \leq \frac{2(\langle \boldsymbol{x}, \boldsymbol{w} \rangle + b)}{n - b} + 1\right\}. \quad (11)$$

Hence, for each layer $i \in [\![1, L]\!]$, we can replace the quadratic function $\boldsymbol{g}_i$ by two linear functions

$$\begin{cases} \boldsymbol{g}_{i,\text{LIN}}^1(\boldsymbol{x}_i, \boldsymbol{x}_{i-1}) := (\text{nv}(\boldsymbol{W}^{[i]}) + \boldsymbol{b}^{[i]}) \odot (\boldsymbol{x}_i + \boldsymbol{1}) - 2\left(\boldsymbol{W}^{[i]}\boldsymbol{x}_{i-1} + \boldsymbol{b}^{[i]}\right), & (12a) \\ \\ \boldsymbol{g}_{i,\text{LIN}}^2(\boldsymbol{x}_i, \boldsymbol{x}_{i-1}) := (\text{nv}(\boldsymbol{W}^{[i]}) - \boldsymbol{b}^{[i]}) \odot (\boldsymbol{1} - \boldsymbol{x}_i) + 2\left(\boldsymbol{W}^{[i]}\boldsymbol{x}_{i-1} + \boldsymbol{b}^{[i]}\right), & (12b) \end{cases}$$

Thus, by encoding the $\text{sign}(\cdot)$ function as described in (12a)-(12b), the standard BNN verification problem can be equivalently formulated as an instance of MILP:

$$\tau_{\text{MILP}} := \begin{cases} \min\limits_{\boldsymbol{x}_0, \boldsymbol{x}_1, \ldots, \boldsymbol{x}_L} f(\boldsymbol{x}_0, \boldsymbol{x}_1, \ldots, \boldsymbol{x}_L) \\ \text{s.t.} \quad \boldsymbol{g}_{i,\text{LIN}}^1(\boldsymbol{x}_i, \boldsymbol{x}_{i-1}) \geq \boldsymbol{0}, \ i \in [\![1, L]\!], & (13a) \\ \quad\quad \boldsymbol{g}_{i,\text{LIN}}^2(\boldsymbol{x}_i, \boldsymbol{x}_{i-1}) \geq \boldsymbol{0}, \ i \in [\![1, L]\!], & (13b) \\ \quad\quad \boldsymbol{g}_{\boldsymbol{B}}(\boldsymbol{x}_0) \geq \boldsymbol{0}, & (13c) \\ \quad\quad \boldsymbol{x}_i \in \{-1, 1\}^{n_i}, \ i \in [\![1, L]\!]. & (13d) \end{cases}$$

By further relaxing the binary constraints in (13d) via $\boldsymbol{g}_{i,\text{LIN}}^0 := (\boldsymbol{1} - \boldsymbol{x}_i, \boldsymbol{x}_i + \boldsymbol{1})$, we can derive the corresponding LP relaxation of the MILP problem in (13)[2]:

$$\tau_{\text{LP}} := \begin{cases} \min\limits_{\boldsymbol{x}_0, \boldsymbol{x}_1, \ldots, \boldsymbol{x}_L} f(\boldsymbol{x}_0, \boldsymbol{x}_1, \ldots, \boldsymbol{x}_L) \\ \text{s.t.} \quad \boldsymbol{g}_{i,\text{LIN}}^0(\boldsymbol{x}_i) \geq \boldsymbol{0}, i \in [\![1, L]\!], & (14a) \\ \quad\quad (13a) - (13c). & (14b) \end{cases}$$

---

[2]Replacing the initial sign constraint by $\boldsymbol{g}_{1,\text{LIN}}^1$ and $\boldsymbol{g}_{1,\text{LIN}}^2$ provides a valid encoding only if $\boldsymbol{x}_0 \in [-1, 1]^{n_0}$. However, it is always possible to transform the input perturbation region so that it corresponds to $\mathbb{B}_{||.||_\infty}(\boldsymbol{0}, 1)$.

Notice that we always have $\tau_{\text{LP}} \leq \tau_{\text{MILP}} = \tau$.

**Remark 3.1** (Encoding of $\text{sign}(\cdot)$). *In Lazarus & Kochenderfer (2022); Khalil et al. (2019), the authors have also considered a* MILP *encoding of the* BNN *verification problem, based on $l, u \in \mathbb{R}$ such that $l \leq \langle \boldsymbol{x}, \boldsymbol{w} \rangle + b \leq u$ and*

$$y = \text{sign}(\langle \boldsymbol{x}, \boldsymbol{w} \rangle + b) \implies \frac{4}{u}(\langle \boldsymbol{x}, \boldsymbol{w} \rangle + b) - 3 \leq y \text{ and } y \geq -\frac{4}{l}(\langle \boldsymbol{x}, \boldsymbol{w} \rangle + b) + 1. \qquad (15)$$

*For $\boldsymbol{w} \in \{-1, 1\}^n$ and $u = n + b$, we conclude that (11) provides a tighter bound than (15) because*

$$y - \left( \frac{4}{n+b} (\langle \boldsymbol{x}, \boldsymbol{w} \rangle + b) - 3 \right) = y - \left( \frac{2}{n+b} (\langle \boldsymbol{x}, \boldsymbol{w} \rangle + b) - 1 \right) + \sum_{k=1}^{n} \frac{(1 - w_k x_k)^2}{n + b}. \qquad (16)$$

**Theorem 3.1.** *For an arbitrary* BNN *with depth $L \geq 2$, there always exists an affine function $f : \mathbb{R}[\boldsymbol{x}_0, \boldsymbol{x}_1, \ldots, \boldsymbol{x}_L] \to \mathbb{R}$ such that $\tau_{\text{LP}} > \tau^1 = \tau_{\text{cs}}^1$.*

*Sketch of proof.* Consider $f(\boldsymbol{x}_0, \boldsymbol{x}_1, \ldots, \boldsymbol{x}_L) = \boldsymbol{g}_{L,\text{LIN}}^1(\boldsymbol{x}_L, \boldsymbol{x}_{L-1})_j$ for some $j \in [\![1, n_L]\!]$. Then Appendix A.1 explicitly constructs a feasible solution to the dual of the sparse first-order SDP relaxation of (8), yielding the value of the objective function equal to $-1$, implying $\tau_{\text{cs}}^1 = \tau^1 \leq -1$. Since $f(\boldsymbol{x}_0, \boldsymbol{x}_1, \ldots, \boldsymbol{x}_L) \geq 0$ is one of the constraints in (14), we deduce that $\tau_{\text{LP}} \geq 0$. $\qquad \square$

Theorem 3.1 states that the lower bound $\tau^1$ is generally not competitive. On the other hand, the inequality $\tau^2 \geq \tau_{\text{LP}}$ is derived in Appendix A.2, but the second-order SDP relaxation remains computationally inefficient for high-dimensional problems.

## 4 TIGHTENING OF THE FIRST-ORDER SDP RELAXATION

The goal of this section is to propose a more accurate first-order SDP relaxation. We still assume that $\boldsymbol{B} = \mathbb{B}_{||\cdot||_\infty}(\bar{\boldsymbol{x}}, \varepsilon)$. Firstly, notice that the semi-algebraic representation of the subgradient of the ReLU function derived in (Chen et al., 2020, Section 1.3) provides an additional way of exactly encoding the $\text{sign}(\cdot)$ function using quadratic polynomials. Consequently, for each $i \in [\![1, L]\!]$, let us replace the constraint defined in (7b) by the following two constraints:

$$\begin{cases} \tilde{\boldsymbol{g}}_i^1(\boldsymbol{x}_i, \boldsymbol{x}_{i-1}) := (\boldsymbol{x}_i + \boldsymbol{1}) \odot \left( \boldsymbol{W}^{[i]} \boldsymbol{x}_{i-1} + \boldsymbol{b}^{[i]} \right) \geq \boldsymbol{0}, & (17a) \\[2mm] \tilde{\boldsymbol{g}}_i^2(\boldsymbol{x}_i, \boldsymbol{x}_{i-1}) := (\boldsymbol{x}_i - \boldsymbol{1}) \odot \left( \boldsymbol{W}^{[i]} \boldsymbol{x}_{i-1} + \boldsymbol{b}^{[i]} \right) \geq \boldsymbol{0}. & (17b) \end{cases}$$

Furthermore, we include the following two *redundant* quadratic constraints (tautologies):

$$\begin{cases} \tilde{\boldsymbol{g}}_i^{\text{t1}}(\boldsymbol{x}_i, \boldsymbol{x}_{i-1}) := (\boldsymbol{x}_i + \boldsymbol{1}) \odot \left( \text{nv}\left( \boldsymbol{W}^{[i]} \right) - \boldsymbol{W}^{[i]} \boldsymbol{x}_{i-1} \right) \geq \boldsymbol{0}, & (18a) \\[2mm] \tilde{\boldsymbol{g}}_i^{\text{t2}}(\boldsymbol{x}_i, \boldsymbol{x}_{i-1}) := (\boldsymbol{1} - \boldsymbol{x}_i) \odot \left( \text{nv}\left( \boldsymbol{W}^{[i]} \right) + \boldsymbol{W}^{[i]} \boldsymbol{x}_{i-1} \right) \geq \boldsymbol{0}, & (18b) \end{cases}$$

which hold true for $\boldsymbol{x}_{i-1} \in \{-1, 1\}^{n_{i-1}}$ and $\left| \boldsymbol{W}^{[i]} \boldsymbol{x}_{i-1} \right| \leq \text{nv}\left( \boldsymbol{W}^{[i]} \right)$, where $|\cdot|$ should be understood component-wise.

Hence, we get an alternative POP encoding of the BNN verification problem:

$$\tau_{\text{tighter}} := \begin{cases} \displaystyle\min_{\boldsymbol{x}_0, \boldsymbol{x}_1, \ldots, \boldsymbol{x}_L} f(\boldsymbol{x}_0, \boldsymbol{x}_1, \ldots, \boldsymbol{x}_L) & \\ \text{s.t.} \quad \boldsymbol{h}_i(\boldsymbol{x}_i) = \boldsymbol{0}, \ i \in [\![1, L]\!], & (19a) \\ \quad \tilde{\boldsymbol{g}}_i^1(\boldsymbol{x}_i, \boldsymbol{x}_{i-1}) \geq \boldsymbol{0}, \ \tilde{\boldsymbol{g}}_i^{\text{t1}}(\boldsymbol{x}_i, \boldsymbol{x}_{i-1}) \geq \boldsymbol{0}, \ i \in [\![1, L]\!], & (19b) \\ \quad \tilde{\boldsymbol{g}}_i^2(\boldsymbol{x}_i, \boldsymbol{x}_{i-1}) \geq \boldsymbol{0}, \ \tilde{\boldsymbol{g}}_i^{\text{t2}}(\boldsymbol{x}_i, \boldsymbol{x}_{i-1}) \geq \boldsymbol{0}, \ i \in [\![1, L]\!], & (19c) \\ \quad \boldsymbol{g}_B(\boldsymbol{x}_0) \geq \boldsymbol{0}, & (19d) \end{cases}$$

and corresponding dense and sparse hierarchies of SDP relaxations, i.e., $(\tau_{\text{tighter}}^d)_d$ and $(\tau_{\text{tighter,cs}}^d)_d$.

**Theorem 4.1.** *For any* BNN *verification problem, $\tau_{\text{tighter,cs}}^1 = \tau_{\text{tighter}}^1 \geq \tau^1$. If $L \geq 2$, there exists an affine $f$ such that the inequality is strict. We also have $\tau_{\text{tighter,cs}}^1 \geq \tau_{\text{LP}}$ for any $f$ affine.*

Theorem 4.1 asserts that adding tautologies (18a) and (18b) is crucial, as they allow to generate a larger first-order quadratic module and consequently enable more accurate SOS decompositions. See Appendix A.3 for the proof and experimental illustration.

**Illustration for the case** $L = 2$ (refer to Appendix A.4 for the general case)**:** Generally, the BNN structure allows us to decompose the problem by considering $n_0 + n_2$ subsets of variables, each of size $n_1 + 1$. Those subsets are given by $I_k = \{\boldsymbol{x}_{1,1}, \ldots, \boldsymbol{x}_{1,n_1}, \boldsymbol{x}_{0,k}\}$ for $k \in [\![1, n_0]\!]$ and $I_{n_0+k} = \{\boldsymbol{x}_{1,1}, \ldots, \boldsymbol{x}_{1,n_1}, \boldsymbol{x}_{2,k}\}$ for $k \in [\![1, n_2]\!]$. [3]

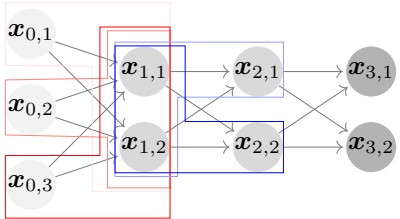

Figure 1: A toy BNN with $L = 2$ and $(n_0, n_1, n_2, n_3) = (3, 2, 2, 2)$. The subsets of interacting variables $I_1 = \{x_{0,1}, x_{1,1}, x_{1,2}\}, I_2 = \{x_{0,2}, x_{1,1}, x_{1,2}\}, I_3 = \{x_{0,3}, x_{1,1}, x_{1,2}\}$ (represented by red polygons) and $I_4 = \{x_{1,1}, x_{1,2}, x_{2,1}\}, I_5 = \{x_{1,1}, x_{1,2}, x_{2,2}\}$ (represented by blue polygons) are used to compute $\tau^1_{\text{tighter,cs}}$.

For the BNN from Example 2.1, depicted in Figure 1, most SOS multipliers are non-negative reals when $d = 1$. With $\tilde{\boldsymbol{g}}_i = \{\tilde{\boldsymbol{g}}_i^1, \tilde{\boldsymbol{g}}_i^2, \tilde{\boldsymbol{g}}_i^{t1}, \tilde{\boldsymbol{g}}_i^{t2}\}$, the tighter first-order sparse SDP relaxation writes:

$$\tau^1_{\text{tighter,cs}} = \begin{cases} \sup\limits_{\lambda, \{\boldsymbol{\sigma_g}\}_{\boldsymbol{g} \in \tilde{\boldsymbol{g}}_i, i \in [\![1,2]\!]}, \sigma_{\boldsymbol{B}}, \{\boldsymbol{G}_k\}_{k=1}^5} \lambda \\[2mm] \text{s.t.} \quad f_1^{\text{adv}} - \lambda - \sigma \in \mathcal{I}_1(h), & (20\text{a}) \\[2mm] \quad \sigma(\boldsymbol{x}) = \sum\limits_{i=1}^2 \sum\limits_{\boldsymbol{g} \in \tilde{\boldsymbol{g}}_i} \mathbf{1}^{\mathsf{T}}(\boldsymbol{\sigma_g} \odot \boldsymbol{g}(\boldsymbol{x}_i, \boldsymbol{x}_{i-1})) + \sum\limits_{k=1}^5 \sigma_{0,k}(\boldsymbol{x}_{I_k}) + \sigma_{\boldsymbol{B}} g_{\boldsymbol{B}}(\boldsymbol{x}_0), & (20\text{b}) \\[2mm] \quad \sigma_{0,k}(\boldsymbol{x}_{I_k}) = \boldsymbol{v}_1(\boldsymbol{x}_{I_k})^{\mathsf{T}} \boldsymbol{G}_k \boldsymbol{v}_1(\boldsymbol{x}_{I_k}), \boldsymbol{G}_k \in \mathbb{S}_+^{|I_k|+1}, k \in [\![1, 5]\!], & (20\text{c}) \\[2mm] \quad \boldsymbol{\sigma_g} \geq \boldsymbol{0}, \boldsymbol{g} \in \tilde{\boldsymbol{g}}_i, i \in [\![1, 2]\!], \qquad \sigma_{\boldsymbol{B}} \geq 0. & (20\text{d}) \end{cases}$$

## 5 NUMERICAL EXPERIMENTS

In this section, we provide numerical results for BNN robustness verification problems with respect to different perturbations. All experiments are run on a desktop with a 12-core i7-12700 2.10 GHz CPU and 32GB of RAM. The tightened first-order sparse SDP relaxation is modeled with TSSOS (Magron & Wang, 2021) and solved with Mosek (Andersen & Andersen, 2000). Gurobi (Gurobi Optimization, LLC, 2023) is used to solve MILP. BNNs were trained on standard benchmark datasets, using Larq Geiger & Team (2020). The full experimental setup is detailed in Appendix B.

### 5.1 ROBUSTNESS AGAINST $\|.\|_\infty$ ATTACKS

MNIST dataset was used to train two sparse networks - $\text{BNN}_1$ and $\text{BNN}_2$, where sparsity refers to *weights sparsity* defined by $w_s =: \frac{1 - \sum_{i=0}^L \|\boldsymbol{W}^{[i+1]}\|_F^2}{\sum_{i=0}^L n_i n_{i+1}}$. We assess the performance of our method (number of solved cases (cert.) and verification time $t$ $(s)$) in verifying robustness of the first 100 images from the test set. Obtained results, see Table 1, are compared with both LP (14) and MILP (13) methods, where the latter was converted into an easier, *attack feasibility problem* by adding $f(\cdot) \leq 0$ to the constraint set.

---

[3]Notice that, unlike in Newton & Papachristodoulou (2023), the presented structure of decision variables subsets is not dependent on the input size $n_0$, which enhances its computational efficiency.

Our method is on average $15.75\%$ (for BNN$_1$) and $25.67\%$ (for BNN$_2$) less conservative than the LP-based method. Moreover, larger input regions do not significantly affect performance, compared to MILP, where the impact on running time is much more severe. This is partially due to the coarseness of LP bounds, see Figure 2, and is even more prominent for densely connected networks.

Table 1: Performance comparison for different models and input regions, given by $\delta_{||\cdot||_\infty} = 127.5\epsilon$ (data were scaled to $[-1, 1]^{784}$). We use $|$ to separate the number of MILP-solver-certified non-robust and robust instances, within a time limitation of $600\,s$. The runtime in parentheses refers to the average runtime over the instances that our method verified successfully.

| Model | $\delta_{||\cdot||_\infty}$ | $\tau_{\text{LP}}$ cert. | $t\,(s)$ | $\tau^1_{\text{tighter,cs}}$ cert. | $t\,(s)$ | $\tau_{\text{Soft-MILP}}$ cert. | $t\,(s)$ |
|---|---|---|---|---|---|---|---|
| BNN$_1$: | 0.25 | 83 | 0.01 | 91 | 3.62 (3.58) | 3 \| 95 | 0.04 (0.04) |
| $[784, 500, 500, 10]$ | 0.50 | 31 | 0.02 | 60 | 6.69 (6.50) | 4 \| 94 | 1.21 (0.06) |
| $w_s = 34.34\%$ | 1.00 | 1 | 0.03 | 21 | 10.76 (8.22) | 15 \| 50 | 251.90 (1.37) |
| | 1.50 | 0 | 0.06 | 6 | 38.32 (26.99) | 20 \| 12 | 428.24 (191.95) |
| BNN$_2$: | 0.25 | 14 | 0.03 | 59 | 11.97 (10.64) | 3 \| 95 | 2.23 (0.69) |
| $[784, 500, 500, 10]$ | 0.50 | 0 | 0.05 | 23 | 42.37 (24.21) | 9 \| 63 | 220.53 (13.24) |
| $w_s = 19.07\%$ | 0.75 | 0 | 0.08 | 9 | 139.18 (52.01) | 10 \| 19 | 455.61 (186.54) |

Figure 2: Comparing $\tau_{\text{LP}}$ and $\tau^1_{\text{tighter,cs}}$ bounds for BNN$_1$ and different $\delta_{||\cdot||_\infty}$. Each subplot $x$-axis represents indices of test set images sorted in the descending order of $\tau^1_{\text{tighter,cs}}$ values. The upper bound $\text{ub}$ is obtained by random sampling. The relative improvement over LP is estimated through $\frac{\tau^1_{\text{tighter,cs}} - \tau_{\text{LP}}}{\text{ub} - \tau_{\text{LP}}}$. On average, $\tau^1_{\text{tighter,cs}}$ bounds are $21, 33, 46$ and $53$ percent more accurate, respectively.

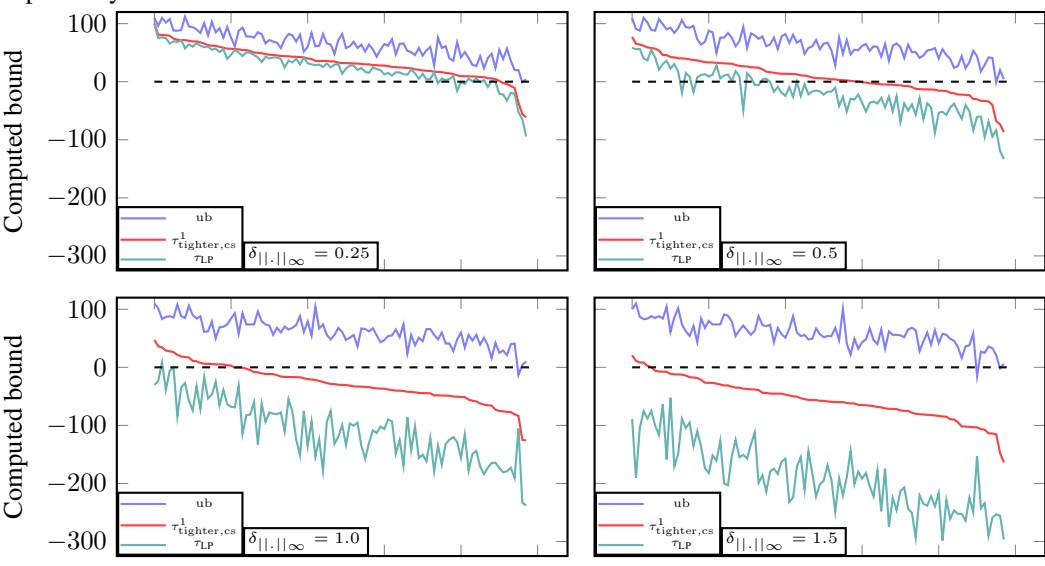

Additional experimental results on $\|\cdot\|_\infty$ robustness verification on CIFAR-10 data set are presented in Appendix B, Table 8.

**SDP methods provide trusted bounds:** The discontinuity of the $\text{sign}(\cdot)$ activation function can exacerbate floating-point errors, leading to significant numerical inaccuracies in the MILP solving process. For instance, if a node's value after linear transformations is $-1.95 \cdot 10^{-14}$, assigning the value of $-1$ to this node is unreliable, as floating-point errors could easily flip the sign. Such incorrect sign values can critically affect the feasibility of the solution and the overall bound. To

address this, we introduce a margin of $10^{-7}$ for sign determination, which we call Soft-MILP. In contrast, for the SDP-based method, we can easily enclose the errors with interval arithmetic as in (Magron et al., 2015, Section 2.2), to obtain a rigorously valid lower bound.

## 5.2 ROBUSTNESS AGAINST $||.||_2$ ATTACKS

We replace the constraint (13c) with $g_B$ such that $B = \mathbb{B}_{||.||_2}(\bar{x}, \varepsilon) \cap [-1, 1]^{784}$. The resulting Mixed Integer Non-Linear Programming (MINP) problem, also solved using Gurobi, and its optimal value are referred to as $\tau_{\text{Soft-MINP}}$.

Table 2: Performance comparison for $||.||_2$-verification, where $\delta_{||.||_2} = 255\varepsilon$.

| $\delta_{||.||_2}$ | $\tau^1_{\text{tighter,cs}}$ | | $\tau_{\text{Soft-MINP}}$ | |
|---|---|---|---|---|
| | cert. | $t\,(s)$ | cert. | $t\,(s)$ |
| BNN$_1$ : $[784, 500, 500, 10], w_s = 34.34\%$ | | | | |
| 10 | 70 | 5.23 (5.02) | 3 \| 93 | 33.35 (5.75) |
| 20 | 36 | 19.54 (15.20) | 4 \| 30 | 447.11 (278.38) |
| 30 | 13 | 34.24 (18.08) | 4 \| 6 | 556.07 (467.16) |
| BNN$_2$ : $[784, 500, 500, 10], w_s = 19.07\%$ | | | | |
| 5 | 81 | 8.57 (8.50) | 2 \| 96 | 3.31 (1.18) |
| 10 | 46 | 19.00 (15.44) | 3 \| 58 | 272.92 (106.25) |
| 15 | 27 | 63.21 (36.73) | 4 \| 23 | 475.78 (293.96) |

As displayed in Table 2, when the perturbation region is small, the exact MINP method can verify more instances. However, for more severe attacks, our method certifies robustness for almost the same number of instances, but $11.4$ times faster on average (for more details, refer to Appendix B).

## 6 CONCLUSION AND FUTURE WORKS

In this work, we studied SDP relaxations associated with polynomial optimization problems to verify the properties of BNNs with continuous input space. We demonstrated the ability of our method to verify robustness against both $||.||_\infty$ and $||.||_2$ attacks. The proposed method efficiently exploits the inherent sparse structure of a given BNN and generally provides much less conservative bounds than the LP-based method. Moreover, its running time does not scale exponentially with either the size of the network or the perturbation region.

Our method relies on the interior-point SDP solvers, and thus inherits their limitations. One direction for future improvements could be based on the embedding of automatic differentiation to either improve existing interior-point SDP solvers or create new ones, like in Dathathri et al. (2020).

Moreover, our experimental results suggest that one could replace LP relaxations with SDP relaxations in branch-and-bound/branch-and-cut algorithms, such as the ones implemented in Gurobi. Based on our experiments, an alternative SDP-based relaxation embedded within general-purpose MILP/MINP solvers would significantly accelerate the exact verification process, especially for larger input perturbation regions.

Better bounding at comparable time cost could significantly improve the overall performance of such solvers, which is another exciting topic of future research.

ACKNOWLEDGMENTS

This work benefited from the HORIZON–MSCA-2023-DN-JD of the European Commission under the Grant Agreement No 101120296 (TENORS), the ANITI AI Cluster program under the Grant agreement n° ANR-23-IACL-0002, the Basic Science Center Program (No: 12288201) of the National Natural Science Foundation of China, the National Key R&D Program of China (2023YFA1009401), as well as the National Research Foundation, Prime Minister's Office, Singapore under its Campus for Research Excellence and Technological Enterprise (CREATE) programme.

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

## A  THEORETICAL DISCUSSION

### A.1  DUAL SIDE OF POLYNOMIAL OPTIMIZATION RELAXATIONS - MOMENT HIERARCHIES

We have discussed computing a global minimum of a multivariate polynomial using SDP relaxations obtained by interpreting the requirement for polynomials to be positive on sets defined with finitely many (in)equalities.

It is also possible to derive the corresponding dual relaxations. Notice that a polynomial $f$ in the variable $\boldsymbol{x} = (x_1, \ldots, x_n)$ can also be written as $f = \sum_{\alpha \in \mathcal{A}} f_\alpha \boldsymbol{x}^\alpha$ with $\mathcal{A} \subset \mathbb{N}^n$ and $f_\alpha \in \mathbb{R}$, where $\boldsymbol{x}^\alpha = x_1^{\alpha_1} \ldots x_n^{\alpha_n}$. Then the *moment hierarchy* (Lasserre, 2001) for the POP introduced in Section 2.3 corresponds to:

$$
\tau_{\text{mom}}^d := \begin{cases}
\inf_{\boldsymbol{y}} L_{\boldsymbol{y}}(f) \\
\text{s.t.} \quad \boldsymbol{M}_d(\boldsymbol{y}) \succeq 0, & \text{(21a)} \\
\qquad \boldsymbol{M}_{d-1}(\boldsymbol{g}_i(\cdot)_j \boldsymbol{y}) \succeq 0, i \in [\![1, L]\!], j \in [\![1, n_i]\!], & \text{(21b)} \\
\qquad \boldsymbol{M}_{d-1}(\boldsymbol{h}_i(\cdot)_j \boldsymbol{y}) = 0, i \in [\![1, L]\!], j \in [\![1, n_i]\!], & \text{(21c)} \\
\qquad \boldsymbol{M}_{d-1}(\boldsymbol{g}_{\boldsymbol{B}}(\cdot)_k \boldsymbol{y}) \succeq 0, k \in [\![1, n_{\boldsymbol{B}}]\!], & \text{(21d)} \\
\qquad y_0 = 1, & \text{(21e)}
\end{cases}
$$

where $\boldsymbol{y} = (y_\alpha)_\alpha$ is a sequence indexed by $\alpha \in \mathbb{N}^n$ and $L_{\boldsymbol{y}}$ the linear functional defined by

$$
f \mapsto L_{\boldsymbol{y}}(f) := \sum_\alpha f_\alpha y_\alpha. \tag{22}
$$

For $d \in \mathbb{N}$, $\boldsymbol{M}_d(\boldsymbol{y})$ denotes the *moment matrix* of order $d$ associated with $\boldsymbol{y}$ and defined as follows

$$
\boldsymbol{M}_d(\boldsymbol{y})(\beta, \gamma) := L_{\boldsymbol{y}}(\boldsymbol{x}^\beta \boldsymbol{x}^\gamma) = y_{\beta+\gamma}, \quad \forall \beta, \gamma \in \mathbb{N}_d^n. \tag{23}
$$

Similarly, for $g = \sum_\alpha g_\alpha \boldsymbol{x}^\alpha \in \mathbb{R}[\boldsymbol{x}]$, $\boldsymbol{M}_d(g\boldsymbol{y})$ denotes the *localizing matrix* of order $d$ associated with $g$ and $\boldsymbol{y}$, defined as follows

$$
\boldsymbol{M}_d(g\boldsymbol{y})(\beta, \gamma) := L_{\boldsymbol{y}}(g\boldsymbol{x}^\beta \boldsymbol{x}^\gamma) = \sum_\alpha g_\alpha y_{\alpha+\beta+\gamma}, \quad \forall \beta, \gamma \in \mathbb{N}_d^n. \tag{24}
$$

When the correlative sparsity is present, analogous hierarchies of sparse moment relaxations can be derived as well (Lasserre, 2006; Magron & Wang, 2023). Such reasoning was adopted in the following proof of Theorem 3.1.

*Proof of Theorem 3.1.* Let $j \in [\![1, n_L]\!]$ and $c_j^L = \text{nv}(\boldsymbol{W}^{[L]})_j + \boldsymbol{b}_j^{[L]}$ and define $f(\boldsymbol{x}_0, \boldsymbol{x}_1, \ldots, \boldsymbol{x}_L) := \boldsymbol{x}_{L,j} - \left( \frac{2}{c_j^L} \left( \left\langle \boldsymbol{W}_{(j,:)}^{[L]}, \boldsymbol{x}_{L-1} \right\rangle + \boldsymbol{b}_j^{[L]} \right) - 1 \right)$. Then,

$$
\boldsymbol{M} = \begin{array}{c} \\ 1 \\ \boldsymbol{x}_{L-1} \\ \boldsymbol{x}_{L,j} \end{array} \begin{array}{c} \overset{1}{\phantom{x}} \quad \overset{\boldsymbol{x}_{L-1}}{\phantom{x}} \quad \overset{\boldsymbol{x}_{L,j}}{\phantom{x}} \\ \begin{bmatrix} 1 & \left( \boldsymbol{W}_{(j,:)}^{[L]} \right)^\mathsf{T} & 0 \\ \boldsymbol{W}_{(j,:)}^{[L]} & \boldsymbol{W}_{(j,:)}^{[L]} \left( \boldsymbol{W}_{(j,:)}^{[L]} \right)^\mathsf{T} & \boldsymbol{0} \\ 0 & \boldsymbol{0} & 1 \end{bmatrix} \end{array} \tag{25}
$$

represents a feasible moment matrix for the sparse first-order moment relaxation of (8), yielding the value of the objective function equal to $-1$ (because $\|\boldsymbol{W}_{(j,:)}^{[L]}\|_2^2 = \text{nv}(\boldsymbol{W}^{[L]})_j$), implying that $\tau^1 \leq -1$. However, since $f(\boldsymbol{x}_0, \boldsymbol{x}_1, \ldots, \boldsymbol{x}_L) \geq 0$ is one of the constraints in (14), we deduce that $\tau_{\text{LP}} \geq 0$.  $\square$

As a direct consequence of Theorem 3.1, we have the following corollary:

**Corollary A.1.1.** *Consider an arbitrary BNN with depth $L \geq 2$, and let $j \in [\![1, n_L]\!]$. Then,*

$$
\boldsymbol{g}_{L,LIN}^1(\cdot)_j \notin \mathcal{Q}_1(\{\boldsymbol{g}_L(\cdot)_j\}), \tag{26}
$$

*thus the set $\{\boldsymbol{g}_{i,LIN}^0, \boldsymbol{g}_{i,LIN}^1, \boldsymbol{g}_{i,LIN}^2, i \in [\![1, L]\!]\}$ is not included in $\mathcal{Q}_1(\{\boldsymbol{g}_1, \ldots, \boldsymbol{g}_L, \boldsymbol{g}_{\boldsymbol{B}}\})$.*

## A.2 Bounds from the second-order relaxation

**Theorem A.2.1.** *If $(\boldsymbol{x}_0, \boldsymbol{x}_1, \ldots, \boldsymbol{x}_L) \mapsto f(\boldsymbol{x}_0, \boldsymbol{x}_1, \ldots, \boldsymbol{x}_L)$ is affine, we have $\tau \geq \tau^2 \geq \tau_{\mathrm{LP}}$. Consequently, any dual-feasible solution of (14) yields a valid SOS decomposition for the second-order SDP relaxation of (8).*

*Proof of Theorem A.2.1.* We proceed by analyzing the feasible sets of each problem. Firstly, both problems share the same input perturbation region given by (13c). For the remaining constraints, let us consider three different cases:

*(i) Single node bounds:* Let $i \in [\![1, L]\!]$ and $j \in [\![1, n_i]\!]$. Then, the following equations on $\mathbb{R}[\boldsymbol{x}_i]$ hold true:

$$\boldsymbol{g}_{i,\mathrm{LIN}}^0(\boldsymbol{x}_i)_j = \frac{1}{2}(1 - \boldsymbol{x}_{i,j})^2 + \frac{1}{2}\boldsymbol{h}_i(\boldsymbol{x}_i)_j, \text{ and } \boldsymbol{g}_{i,\mathrm{LIN}}^0(\boldsymbol{x}_i)_{n_i+j} = \frac{1}{2}(\boldsymbol{x}_{i,j} + 1)^2 + \frac{1}{2}\boldsymbol{h}_i(\boldsymbol{x}_i)_j. \quad (27)$$

Thus $\{\boldsymbol{g}_{i,\mathrm{LIN}}^0, i \in [\![1, L]\!]\} \subset \mathcal{Q}_2(\{\boldsymbol{g}_1, \ldots, \boldsymbol{g}_L, \boldsymbol{g}_B\})$.

*(ii) Interaction between two adjacent hidden layers:* Fix $i \in [\![2, L]\!]$ and $j \in [\![1, n_i]\!]$. Let us define $c_{\pm,j}^i := \mathrm{nv}(\boldsymbol{W}^{[i]})_j \pm \boldsymbol{b}_j^{[i]} > 0$. If we apply the substitution rules $\boldsymbol{x}_{i-1} \odot \boldsymbol{x}_{i-1} = \boldsymbol{1}$ and $\boldsymbol{x}_{i,j}^2 = 1$, then the following equations always hold true on $\mathbb{R}[\boldsymbol{x}_{i-1}, \boldsymbol{x}_{i,j}]$:

$$\boldsymbol{g}_{i,\mathrm{LIN}}^1(\boldsymbol{x}_i, \boldsymbol{x}_{i-1})_j = \frac{(1 - \boldsymbol{x}_{i,j})^2}{2c_{+,j}^i}\boldsymbol{g}_i(\boldsymbol{x}_i, \boldsymbol{x}_{i-1})_j + \frac{(1 + \boldsymbol{x}_{i,j})^2}{4c_{+,j}^i}\sum_{k=1}^{n_{i-1}}\left(1 - \boldsymbol{W}_{(j,k)}^{[i]}\boldsymbol{x}_{i-1,k}\right)^2, \quad (28)$$

$$\boldsymbol{g}_{i,\mathrm{LIN}}^2(\boldsymbol{x}_i, \boldsymbol{x}_{i-1})_j = \frac{(1 + \boldsymbol{x}_{i,j})^2}{2c_{-,j}^i}\boldsymbol{g}_i(\boldsymbol{x}_i, \boldsymbol{x}_{i-1})_j + \frac{(1 - \boldsymbol{x}_{i,j})^2}{4c_{-,j}^i}\sum_{k=1}^{n_{i-1}}\left(1 + \boldsymbol{W}_{(j,k)}^{[i]}\boldsymbol{x}_{i-1,k}\right)^2. \quad (29)$$

Equations (28) and (29) certify that $\{\boldsymbol{g}_{i,\mathrm{LIN}}^1, \boldsymbol{g}_{i,\mathrm{LIN}}^2, i \in [\![1, L]\!]\} \subset \mathcal{Q}_2(\{\boldsymbol{g}_1, \ldots, \boldsymbol{g}_L, \boldsymbol{g}_B\})$.

*(iii) Interaction between the input layer and the first hidden layer:* For any $j \in [\![1, n_1]\!]$, let us define $c_{\pm,j}^1 := \mathrm{nv}(\boldsymbol{W}^{[1]})_j \pm \boldsymbol{b}_j^{[1]} > 0$. It follows that

$$\boldsymbol{g}_{1,\mathrm{LIN}}^1(\boldsymbol{x}_1, \boldsymbol{x}_0)_j = \frac{(1 - \boldsymbol{x}_{1,j})^2}{2c_{+,j}^1}\boldsymbol{g}_1(\boldsymbol{x}_0, \boldsymbol{x}_1)_j + \frac{(1 + \boldsymbol{x}_{1,j})^2}{2c_{+,j}^1}\sum_{k=1}^{n_0}\left(\left|\boldsymbol{W}_{(j,k)}^{[1]}\right| - \boldsymbol{W}_{(j,k)}^{[1]}\boldsymbol{x}_{0,k}\right), \quad (30)$$

$$\boldsymbol{g}_{1,\mathrm{LIN}}^2(\boldsymbol{x}_1, \boldsymbol{x}_0)_j = \frac{(1 + \boldsymbol{x}_{1,j})^2}{2c_{-,j}^1}\boldsymbol{g}_1(\boldsymbol{x}_0, \boldsymbol{x}_1)_j + \frac{(1 - \boldsymbol{x}_{1,j})^2}{2c_{-,j}^1}\sum_{k=1}^{n_0}\left(\left|\boldsymbol{W}_{(j,k)}^{[1]}\right| + \boldsymbol{W}_{(j,k)}^{[1]}\boldsymbol{x}_{0,k}\right), \quad (31)$$

provide the $\mathcal{Q}_2(\{\boldsymbol{g}_1(\cdot)_j\})$-based representations for $\boldsymbol{g}_{1,\mathrm{LIN}}^1(\cdot)_j$ and $\boldsymbol{g}_{1,\mathrm{LIN}}^2(\cdot)_j$, valid after substituting $\boldsymbol{x}_{1,j}^2 = 1$. $\qquad\square$

**Example A.2.1** (Illustration of Theorem A.2.1)**.** *This theorem indicates that any dual-feasible solution of (14) provides a feasible solution for the second-order SOS relaxation (10) of QCQP (8). For instance, let us consider the toy BNN from Example (2.1). Recall that we aim to minimize the linear function $f_1^{\mathrm{adv}}(\boldsymbol{x}) = 2\boldsymbol{x}_{2,2} - 1$. Since the LP relaxation problem has following constraints:*

$$-\frac{4\boldsymbol{x}_{1,2}}{3} + \frac{4\boldsymbol{x}_{1,1}}{3} + \boldsymbol{x}_{2,2} \geq -\frac{5}{3}, \; -\boldsymbol{x}_{1,1} \geq -1, \; \boldsymbol{x}_{1,2} \geq -1, \; \boldsymbol{x}_{2,2} \geq -1, \quad (32)$$

*the expression*

$$
\begin{aligned}
(2\boldsymbol{x}_{2,2} - 1) + 4 = {} & \frac{3}{10}\left(-\frac{4\boldsymbol{x}_{1,2}}{3} + \frac{4\boldsymbol{x}_{1,1}}{3} + \boldsymbol{x}_{2,2} + \frac{5}{3}\right) + \frac{2}{5}\left(-\boldsymbol{x}_{1,1} + 1\right) \\
& + \frac{2}{5}\left(-\boldsymbol{x}_{1,2} + 1\right) + \frac{17}{10}\left(\boldsymbol{x}_{2,2} + 1\right) \geq 0
\end{aligned}
\quad (33)
$$

*gives one feasible dual solution of the LP relaxation problem. From this dual solution, we can recover the following SOS decomposition, certifying that $f_1^{\mathrm{adv}} \geq -3$:*

$$
\begin{aligned}
(2\boldsymbol{x}_{2,2} - 1) + 4 = {} & \frac{3}{10}\left(\frac{2}{3}(\boldsymbol{x}_{2,2} - 1)\left(\boldsymbol{x}_{1,2} - \boldsymbol{x}_{1,1} - \frac{1}{2}\right) + \frac{2}{3}(1 + \boldsymbol{x}_{2,2})(2 - \boldsymbol{x}_{1,2} + \boldsymbol{x}_{1,1})\right) \\
& + \frac{2}{5} \cdot \frac{1}{2}\left(1 - \boldsymbol{x}_{1,1}\right)^2 + \frac{2}{5} \cdot \frac{1}{2}\left(1 + \boldsymbol{x}_{1,2}\right)^2 + \frac{17}{10} \cdot \frac{1}{2}\left(1 + \boldsymbol{x}_{2,2}\right)^2.
\end{aligned}
\quad (34)
$$

A.3   IMPORTANCE OF TAUTOLOGIES

*Proof of Theorem 4.1.* Since all the variables are bounded, we have $\tau^1_{\text{tighter,cs}} = \tau^1_{\text{tighter}}$. Moreover, the inequality $\tau^1_{\text{tighter,cs}} \geq \tau^1$ is a consequence of the identity

$$g_i(\boldsymbol{x}_i, \boldsymbol{x}_{i-1}) = \frac{1}{2}\tilde{g}^1_i(\boldsymbol{x}_i, \boldsymbol{x}_{i-1}) + \frac{1}{2}\tilde{g}^2_i(\boldsymbol{x}_i, \boldsymbol{x}_{i-1}), \; i \in [\![2, L]\!]. \tag{35}$$

In order to prove that $\tau^1_{\text{tighter,cs}} \geq \tau_{\text{LP}}$ holds, we show that all linear functions $g^1_{i,\text{LIN}}, g^2_{i,\text{LIN}}, i \in [\![1, L]\!]$, involved in the constraints (13a) and (13b) belong to $\mathcal{Q}_1(\{\tilde{g}^1_i, \tilde{g}^2_i, \tilde{g}^{\text{t1}}_i, \tilde{g}^{\text{t2}}_i, i \in [\![1, L]\!]\})$. We consider two different scenarios:

*(i)* Let $i \in [\![2, L]\!]$, $j \in [\![1, n_i]\!]$ and $c^i_{\pm,j} := \text{nv}(\boldsymbol{W}^{[i]})_j \pm \boldsymbol{b}^{[i]}_j > 0$. When we apply the substitution rules $\boldsymbol{x}_{i-1} \odot \boldsymbol{x}_{i-1} = \mathbf{1}$ and $\boldsymbol{x}^2_{i,j} = 1$, then the following equations always hold true on $\mathbb{R}[\boldsymbol{x}_{i-1}, \boldsymbol{x}_{i,j}]$:

$$g^1_{i,\text{LIN}}(\boldsymbol{x}_i, \boldsymbol{x}_{i-1})_j = \frac{1}{c^i_{+,j}}\tilde{g}^2_i(\boldsymbol{x}_i, \boldsymbol{x}_{i-1})_j + \frac{1}{c^i_{+,j}}\tilde{g}^{\text{t1}}_i(\boldsymbol{x}_i, \boldsymbol{x}_{i-1})_j, \tag{36}$$

$$g^2_{i,\text{LIN}}(\boldsymbol{x}_i, \boldsymbol{x}_{i-1})_j = \frac{1}{c^i_{-,j}}\tilde{g}^1_i(\boldsymbol{x}_i, \boldsymbol{x}_{i-1})_j + \frac{1}{c^i_{-,j}}\tilde{g}^{\text{t2}}_i(\boldsymbol{x}_i, \boldsymbol{x}_{i-1})_j. \tag{37}$$

*(ii)* Secondly, for any $j \in [\![1, n_1]\!]$, and $c^1_{\pm,j} = \text{nv}(\boldsymbol{W}^{[1]})_j \pm \boldsymbol{b}^{[1]}_j > 0$, substituting $\boldsymbol{x}^2_{1,j} = 1$ yields the following equalities on $\mathbb{R}[\boldsymbol{x}_0, \boldsymbol{x}_{1,j}]$:

$$g^1_{1,\text{LIN}}(\boldsymbol{x}_1, \boldsymbol{x}_0)_j = \frac{1}{c^1_{+,j}}g_2(\boldsymbol{x}_1, \boldsymbol{x}_0)_j + \frac{1}{c^1_{+,j}}\tilde{g}^{\text{t2}}_1(\boldsymbol{x}_1, \boldsymbol{x}_0)_j, \tag{38}$$

$$g^2_{1,\text{LIN}}(\boldsymbol{x}_1, \boldsymbol{x}_0)_j = \frac{1}{c^1_{+,j}}g_1(\boldsymbol{x}_1, \boldsymbol{x}_0)_j + \frac{1}{c^1_{+,j}}\tilde{g}^{\text{t1}}_1(\boldsymbol{x}_1, \boldsymbol{x}_0)_j, \tag{39}$$

certifying that $g^1_{1,\text{LIN}}(\boldsymbol{x}_1, \boldsymbol{x}_0)_j$ and $g^2_{1,\text{LIN}}(\boldsymbol{x}_1, \boldsymbol{x}_0)_j$ belong to $\mathcal{Q}_1(\{\tilde{g}^1_i, \tilde{g}^2_i, \tilde{g}^{\text{t1}}_i, \tilde{g}^{\text{t2}}_i, i \in [\![1, L]\!]\})$.

Finally, Theorem 3.1 implies that there exists an affine $f$ such that $\tau^1_{\text{tighter,cs}} \geq \tau_{\text{LP}} > \tau^1$.

$\square$

**Remark A.3.1.** *We insist that tautologies (18a) and (18b) are both important and necessary. If we consider the optimization problem without tautologies, where only (17a) and (17b) are used to replace (7b), then we have the following strict containment relationship:*

$$\mathcal{Q}_1(\{\boldsymbol{g_B}, \tilde{g}^1_i, \tilde{g}^2_i, i \in [\![1, L]\!]\}) \subsetneq_{\not\supseteq} \mathcal{Q}_1(\{\boldsymbol{g_B}, \tilde{g}^1_i, \tilde{g}^2_i, \tilde{g}^{t1}_i, \tilde{g}^{t2}_i, i \in [\![1, L]\!]\}). \tag{40}$$

*Indeed, following the same reasoning as in the proof of Theorem 3.1, we obtain that*

$$\boldsymbol{M} = \begin{array}{c} \\ \boldsymbol{x}_{L-1} \\ \boldsymbol{x}_{L,j} \end{array} \overset{\displaystyle \begin{array}{ccc} 1 & \boldsymbol{x}_{L-1} & \boldsymbol{x}_{L,j} \end{array}}{\left[ \begin{array}{ccc} 1 & a\left(\boldsymbol{W}^{[L]}_{(j,:)}\right)^\intercal & 0 \\ a\boldsymbol{W}^{[L]}_{(j,:)} & \boldsymbol{W}^{[L]}_{(j,:)}\left(\boldsymbol{W}^{[L]}_{(j,:)}\right)^\intercal & t\boldsymbol{W}^{[L]}_{(j,:)} \\ 0 & t\left(\boldsymbol{W}^{[L]}_{(j,:)}\right)^\intercal & 1 \end{array} \right]} \tag{41}$$

*with $a = \frac{1}{2}\sqrt{2 - \frac{\left(\boldsymbol{b}^{[L]}_j\right)^2}{\left(\text{nv}(\boldsymbol{W}^{[L]})_j\right)^2}} - \frac{\boldsymbol{b}^{[L]}_j}{2\,\text{nv}(\boldsymbol{W}^{[L]})_j}$ and $t = \sqrt{1-a^2}$ is a feasible moment matrix for*

*which the value of the objective function is $-\frac{\text{nv}(\boldsymbol{W}^{[L]})_j}{\text{nv}(\boldsymbol{W}^{[L]})_j + \boldsymbol{b}^{[L]}_j}\left(\sqrt{2 - \frac{\left(\boldsymbol{b}^{[L]}_j\right)^2}{\left(\text{nv}(\boldsymbol{W}^{[L]})_j\right)^2}} - 1\right) < 0$. To*

*prove that $\boldsymbol{M}$ is positive semidefinite, we observe that $M$ has a $n_{L-1}$-dimensional null space corresponding to the direct sum of the sets $A$ and $B$, where*

$$\begin{aligned} A &:= \left\{(0, \boldsymbol{v}, 0)^\intercal \in \mathbb{R}^{n_{L-1}+2} \mid \left\langle \boldsymbol{W}^{[L]}_{(j,:)}, \boldsymbol{v} \right\rangle = 0\right\}, \\ B &:= \left\{\alpha\left(-a\,\text{nv}(\boldsymbol{W}^{[L]})_j, \boldsymbol{W}^{[L]}_{(j,:)}, -t\,\text{nv}(\boldsymbol{W}^{[L]})_j\right)^\intercal \mid \alpha \in \mathbb{R}\right\}. \end{aligned} \tag{42}$$

*Thus, $M$ has $n_{L-1}$ eigenvalues equal to zero, and one eigenvalue equal to $1$ corresponding to the eigenvector $(-t, \mathbf{0}, a)^\intercal$. Since the trace of $M$ is $\mathrm{nv}(W^{[L]})_j + 2$, we know that the maximal eigenvalue of $M$ is $\mathrm{nv}(W^{[L]})_j + 1$.*

Details from the previous discussion can be summarized in the following corollary:

**Corollary A.3.1.** *Let $i \in [\![2, L]\!]$ and $j \in [\![1, n_i]\!]$. If the substitution rule $x_{i,j}^2 = 1$ is applied on the space $\mathbb{R}[x_{i-1}, x_{i,j}]$,*

$$g_{i,LIN}^1(\cdot)_j \notin \mathcal{Q}_1(\{g_i(\cdot)_j, g_{i,LIN}^0(\cdot)_j, g_{i,LIN}^0(\cdot)_{n_i+j}\}). \tag{43}$$

*Moreover, at least one of the polynomials defining the constraints from (17a)-(18b) is not in $\mathcal{Q}_1(\{g_i(\cdot)_j, g_{i,LIN}^0(\cdot)_j, g_{i,LIN}^0(\cdot)_{n_i+j}\})$, and consequently*

$$\mathcal{Q}_1(\{g_i(\cdot)_j, g_{i,LIN}^0(\cdot)_j, g_{i,LIN}^0(\cdot)_{n_i+j}\}) \subsetneq \mathcal{Q}_1(\{\tilde{g}_i^1, \tilde{g}_i^2, \tilde{g}_i^{tl}, \tilde{g}_i^{t2}, i \in [\![1, L]\!]\}). \tag{44}$$

Numerical experiments, simultaneously illustrating both scalability and bound superiority of $\tau_{\text{tighter,cs}}^1$, are reported in Table 3.

Table 3: Behaviour of bounds from the first-order sparse relaxations on randomly generated $\|.\|_\infty$-verification instances. Parameter $s$ represents row sparsity, i.e., the number of non-zero elements in each of the weight matrices rows, while "dense" indicates fully populated weight matrices. The relative $\tau_{\text{LP}}$ running time being significantly smaller, we omit it from the table.

| Network size and sparsity | bound | | | $t$ $(s)$ | |
|---|---|---|---|---|---|
| | $\tau_{\text{tighter,cs}}^1$ | $\tau_{\text{QCQP}}^1$ | $\tau_{\text{LP}}$ | $\tau_{\text{tighter,cs}}^1$ | $\tau_{\text{QCQP}}^1$ |
| $[300, 100, 100]$, dense | **-24.71** | -78.64 | -78.83 | 413.51 | 427.28 |
| $[500, 100, 100]$, dense | **-27.17** | -78.58 | -77.89 | 795.41 | 721.81 |
| $[300, 100, 100, 100]$, dense | **-19.49** | -77.06 | -76.81 | 739.21 | 992.66 |
| $[500, 100, 100, 100]$, dense | **-32.35** | -82.35 | -81.94 | 928.16 | 964.56 |
| $[1000, 100, 100, 100]$, dense | **-26.19** | -81.70 | -81.32 | 1764.67 | 1659.13 |
| $[300, 100, 100, 100, 100]$, dense | **-30.58** | -80.59 | -80.42 | 966.76 | 1523.92 |
| $[500, 100, 100, 100, 100]$, dense | **-16.19** | -77.61 | -77.25 | 1341.33 | 1498.51 |
| $[1000, 100, 100, 100, 100]$, dense | **-18.31** | -76.32 | -75.89 | 1959.30 | 2237.44 |
| $[300, 100, 100, 100, 100, 100]$, dense | **-28.62** | -80.95 | -80.99 | 1012.18 | 1803.31 |
| $[500, 100, 100, 100, 100, 100]$, dense | **-16.04** | -78.10 | -78.12 | 1255.16 | 1641.50 |
| $[1000, 100, 100, 100, 100, 100]$, dense | **-32.54** | -83.93 | -82.84 | 2225.89 | 2866.54 |
| $[500, 100, 100]$, $s = 10$ | **-80.84** | -94.59 | -98.16 | 12.53 | 14.90 |
| $[800, 200, 200]$, $s = 10$ | **-181.52** | -194.96 | -198.97 | 218.72 | 186.77 |
| $[1000, 100, 100]$, $s = 10$ | **-81.13** | -94.72 | -98.18 | 13.44 | 12.55 |
| $[1000, 200, 200]$, $s = 10$ | **-174.13** | -192.79 | -198.61 | 243.87 | 191.24 |
| $[1000, 300, 300]$, $s = 10$ | **-242.60** | -283.95 | -297.99 | 1203.41 | 1093.90 |
| $[1000, 100, 100, 100]$, $s = 10$ | **-72.79** | -92.98 | -97.78 | 69.70 | 77.63 |
| $[1000, 100, 100, 100, 100]$, $s = 10$ | **-81.81** | -95.30 | -98.30 | 327.83 | 319.04 |
| $[1000, 500, 500]$, $s = 3$ | **-476.01** | -495.79 | -499.78 | 520.61 | 502.97 |
| $[2000, 500, 500]$, $s = 3$ | **-466.91** | -492.96 | -499.69 | 238.35 | 225.89 |
| $[2000, 300, 300, 300]$, $s = 3$ | **-283.58** | -296.90 | -299.75 | 469.02 | 309.20 |
| $[2000, 1000, 1000]$, $s = 2$ | **-944.68** | -989.77 | -999.83 | 207.37 | 245.58 |
| $[3000, 1000, 1000]$, $s = 2$ | **-943.68** | -989.65 | -999.83 | 81.96 | 84.25 |
| $[3000, 2000, 1000]$, $s = 2$ | **-982.09** | -996.68 | -999.97 | 662.55 | 648.97 |
| $[2000, 500, 500, 500]$, $s = 2$ | **-484.19** | -497.30 | -499.90 | 144.28 | 146.75 |
| $[3000, 500, 500, 500]$, $s = 2$ | **-480.42** | -496.66 | -499.88 | 137.56 | 159.70 |

As it can be observed in Table 3, the bound $\tau_{\text{tighter,cs}}^1$ is consistently better, across all instances. Moreover, the running time of $\tau_{\text{tighter,cs}}^1$ is relatively stable, even for networks with large number of neurons, akin to small-to-medium-sized convolutional networks.

A.4 ON DETERMINING THE SUBSETS OF DECISION VARIABLES

Let us describe in details how decision variables are structured within the first-order sparse SDP relaxation of the problem (19). We emphasize that this structure is valid only for the first-order sparse SDP relaxations, and regardless of the norm describing the input perturbation region.
We recall that subsets $(I_k)_{k \in [\![1,p]\!]}$ are said to satisfy the RIP property is

$$(I_{k+1} \cap \cup_{j \leq k} I_j) \subset I_i, \text{ for some } i \leq k. \tag{45}$$

The RIP property (45) is one of the essential requirements for the convergence of the sparse SOS-based hierarchy of SDP relaxations (Magron & Wang, 2023, Assumption 3.1).

**Theorem A.4.1.** *The structure of decision variables of the standard BNN robustness verification problem is such that:*

- *If $L = 1$, there are $n_0$ subsets of size $n_1 + 1$, given by $I_k = \{\boldsymbol{x}_{1,1}, \ldots, \boldsymbol{x}_{1,n_1}, \boldsymbol{x}_{0,k}\}, k \in [\![1, n_0]\!]$.*

- *If $L = 2$, there are $n_0 + n_2$ subsets of size $n_1 + 1$, given by $I_k = \{\boldsymbol{x}_{1,1}, \ldots, \boldsymbol{x}_{1,n_1}, \boldsymbol{x}_{0,k}\}$ for $k \in [\![1, n_0]\!]$ and $I_{n_0+k} = \{\boldsymbol{x}_{1,1}, \ldots, \boldsymbol{x}_{1,n_1}, \boldsymbol{x}_{2,k}\}$ for $k \in [\![1, n_2]\!]$.*

- *If $L \geq 3$, there are $L - 2 + n_0 + n_L$ subsets in total, and their maximum size is given by $\max\{n_1 + 1, n_{L-1} + 1, \max_{j \leq L-2}\{n_j + n_{j+1}\}\}$. The subsets are given by $I_k = \{\boldsymbol{x}_{k,1}, \ldots, \boldsymbol{x}_{k,n_k}, \boldsymbol{x}_{k+1,1}, \ldots, \boldsymbol{x}_{k+1,n_{k+1}}\}$ for $k \in [\![1, L - 2]\!]$, $I_{L-2+k} = \{\boldsymbol{x}_{0,k}, \boldsymbol{x}_{1,1}, \ldots, \boldsymbol{x}_{1,n_1}\}$ for $k \in [\![1, n_0]\!]$ and $I_{L-2+n_0+k} = \{\boldsymbol{x}_{L-1,1}, \ldots, \boldsymbol{x}_{L-1,n_{L-1}}, \boldsymbol{x}_{L,k}\}$ for $k \in [\![1, n_L]\!]$.*

*In any of the three cases, the defined subsets satisfy the RIP property.*

*Proof of Theorem A.4.1.* We consider different cases, depending on the depth of the BNN:

- $L = 1$:

  All the nodes from the hidden layer form a separate subset with each individual input node, resulting in $n_0$ subsets of size $n_1 + 1$. Thus, we can set $I_k = \{\boldsymbol{x}_{1,1}, \ldots, \boldsymbol{x}_{1,n_1}, \boldsymbol{x}_{0,k}\}$, for $k \in [\![1, n_0]\!]$. The RIP property is clearly satisfied, as the nodes from the hidden layer appear in each subset.

- $L = 2$:

  All the nodes from the first hidden layer form a separate subset with each individual input node, resulting in $n_0$ subsets of size $n_1 + 1$. Thus, we can set $I_k = \{\boldsymbol{x}_{1,1}, \ldots, \boldsymbol{x}_{1,n_1}, \boldsymbol{x}_{0,k}\}$, for $k \in [\![1, n_0]\!]$. Moreover, they also form a separate subset with each individual node from the second hidden layer, resulting in $n_2$ subsets of size $n_1 + 1$. Thus, we can set $I_{n_0+k} = \{\boldsymbol{x}_{1,1}, \ldots, \boldsymbol{x}_{1,n_1}, \boldsymbol{x}_{2,k}\}$, for $k \in [\![1, n_2]\!]$. The RIP property is satisfied as the first hidden layer nodes appear in each subset.

- $L \geq 3$:

  For every $k \in [\![1, L - 2]\!]$, a pair $(k, k + 1)$ of adjacent hidden layers forms a subset $I_k = \{\boldsymbol{x}_{k,1}, \ldots, \boldsymbol{x}_{k,n_k}, \boldsymbol{x}_{k+1,1}, \ldots, \boldsymbol{x}_{k+1,n_{k+1}}\}$ of size $n_k + n_{k+1}$. Furthermore, all the nodes from the first hidden layer form a separate subset with each individual input node, resulting in $n_0$ subsets of size $n_1 + 1$. Thus, we can set $I_{L-2+k} = \{\boldsymbol{x}_{1,1}, \ldots, \boldsymbol{x}_{1,n_1}, \boldsymbol{x}_{0,k}\}$, for $k \in [\![1, n_0]\!]$. Finally, all the nodes from the penultimate hidden layer form a separate subset with each individual node from the last hidden layer, resulting in $n_L$ subsets of size $n_{L-1} + 1$. Thus, we can set $I_{L-2+n_0+k} = \{\boldsymbol{x}_{L-1,1}, \ldots, \boldsymbol{x}_{L-1,n_{L-1}}, \boldsymbol{x}_{L,k}\}$, for $k \in [\![1, n_L]\!]$.

  Therefore, there are $L - 2 + n_0 + n_L$ subsets in total, and the maximum subset size is given by $\max\{n_1 + 1, n_{L-1} + 1, \max_{j \leq L-2}\{n_j + n_{j+1}\}\}$. Moreover, given this particular ordering (enumeration) of subsets, the RIP property is satisfied. Indeed,

  – If $k \in [\![2, L - 2]\!]$, then

  $$I_k \cap (\cup_{j \leq k-1} I_j) = \{\boldsymbol{x}_{k-1,1}, \ldots, \boldsymbol{x}_{k-1,n_{k-1}}\} \subset I_{k-1}. \tag{46}$$

- If $k \in [\![L-2+1, L-2+n_0]\!]$, then

$$I_k \cap (\cup_{j \leq k-1} I_j) = \{\boldsymbol{x}_{1,1}, \ldots, \boldsymbol{x}_{1,n_1}\} \subset I_1. \tag{47}$$

- If $k \in [\![L-2+n_0+1, L-2+n_0+n_L]\!]$, then

$$I_k \cap (\cup_{j \leq k-1} I_j) = \{\boldsymbol{x}_{L-1,1}, \ldots, \boldsymbol{x}_{L-1,n_{L-1}}\} \subset I_{L-2}. \tag{48}$$

Thus, the RIP property holds in all three cases, which concludes the proof. $\qquad\square$

An alternative structure of node subsets, independent of the depth of the BNN, is to consider $\{\boldsymbol{x}_{i-1,j}, \boldsymbol{x}_{i,k}\}_{1 \leq j \leq n_{i-1}, 1 \leq k \leq n_i}^{1 \leq i \leq L}$. This choice results in $\sum_{i=1}^{L} n_{i-1} \times n_i$ subsets of size two, since every node is coupled with all the other nodes form adjacent layers. The RIP property always holds in this case if subsets are enumerated in an order that mirrors the feed-forward structure of the BNN. Computations are generally faster for this structure of subsets, but the obtained bounds are much coarser.

# B    EXPERIMENTAL SETUP

**Training details:**   All neural networks have been trained on MNIST dataset (data were scaled to belong to $[-1, 1]^{784}$) using Larq (Geiger & Team, 2020), an open-source Python library for training neural networks with quantized (binarized) weights and activation functions.

The training process, lasting 300 epochs, has consisted of minimizing the sparse categorical cross-entropy, using the Adam optimizer (Kingma & Ba, 2014). The learning rate has been handled by the exponential decay learning scheduler, whose initial value has been set to be 0.001. SteTern quantizer with different threshold values has been used to induce different sparsities of the weights matrices. Network parameters have been initialized from a uniform distribution. Batch normalization layers have been added to all but the output layer.
All networks have achieved over $95\%$ accuracy on the test set.

**Optimization details:**   Semidefinite Programming (SDP) problems have been assembled using TSSOS (Wang et al., 2021a), a specialized Julia library for polynomial optimization. Correlative sparsity has been exploited by using the CS="MF" option, which generates an approximately smallest chordal extension of the identified subsets structure. The SDP values have been computed via the interior-point solver Mosek (Andersen & Andersen, 2000), and the SDP solving time was recorded.

Since computing the exact value of the objective function for MILP and MINP problems is highly computationally demanding, we transform those problems into satisfiability problems by setting the objective function to be a constant function equal to zero, and adding $f(\cdot) \leq 0$ to the set of constraints. Infeasibility of the resulting problem provides a certificate of robustness. These problems have then been solved using Gurobi, where the time limit was set to 600 seconds. The upper bound presented in Figure 2 has been determined by randomly sampling $10,000$ points from the uniform distribution over the perturbation region and documenting the lowest objective function value observed at those points.
Our code is available via the following following link: POP4BNN GitHub Repository.

**More detailed experimental results:**   We provide detailed verification results against $||.||_\infty$ (see Table 4 and Table 5) and $||.||_2$ (see Table 6 and Table 7) attacks. Those results illustrate that the solving time of general branch and bound algorithms could be significantly improved if the LP bounds would be replaced by tighter SDP bounds, such as $\tau_{\text{tighter,cs}}^1$. The improvement would be even more noticeable for $||.||_2$ verification.

Finally, due to their increased computational complexity, the experiments from Table 3 were run on a server with a 26-core Intel(R) Xeon(R) Gold 6348 CPU @ 2.60GHz and a RAM of 756GB.

Table 4: Verification against $\|.\|_\infty$-attacks: detailed numerical results for $\text{BNN}_1$ : $[784, 500, 500, 10], w_s = 34.34\%$. For example, image 72 illustrates that MILP methods can not handle severe attacks. Likewise, image 28 confirms the potential benefits of accommodating $\tau^1_{\text{tighter,cs}}$ within the branch and bound algorithms. Note that MILP implementation only solves the attack feasibility problem, since computing exact bounds would result in much more timeouts.

| Image Index | $\tau^1_{\text{tighter,cs}}$ | | $\tau_{\text{Soft-MILP}}$ | |
|:---:|:---:|:---:|:---:|:---:|
| | bound | $t\,(s)$ | bound | $t\,(s)$ |
| $\delta_{\|.\|_\infty} = 1.25$ | | | | |
| 14 | 1.08 | 11.53 | infeasible | 552.99 |
| 24 | 5.82 | 15.43 | timeout | $> 600$ |
| 26 | 0.31 | 7.18 | infeasible | 4.07 |
| 28 | 29.86 | 14.90 | infeasible | 29.08 |
| 31 | 7.54 | 10.08 | infeasible | 11.67 |
| 52 | 12.17 | 7.53 | infeasible | 0.49 |
| 55 | 11.05 | 6.99 | infeasible | 3.33 |
| 57 | 3.23 | 8.69 | infeasible | 6.41 |
| 72 | 29.43 | 9.75 | infeasible | 1.36 |
| 83 | 17.29 | 14.07 | infeasible | 44.62 |
| 100 | 19.99 | 6.64 | infeasible | 0.30 |
| $\delta_{\|.\|_\infty} = 1.50$ | | | | |
| 28 | 11.18 | 70.62 | timeout | $> 600$ |
| 52 | 0.48 | 18.35 | infeasible | 7.18 |
| 55 | 2.25 | 16.33 | infeasible | 7.42 |
| 72 | 20.65 | 28.52 | infeasible | 15.68 |
| 83 | 8.37 | 17.68 | infeasible | 519.23 |
| 100 | 8.32 | 10.39 | infeasible | 2.21 |
| $\delta_{\|.\|_\infty} = 1.75$ | | | | |
| 28 | 0.20 | 42.46 | timeout | $> 600$ |
| 72 | 13.50 | 37.79 | timeout | $> 600$ |

Table 5: Verification against $\|.\|_\infty$-attacks: detailed numerical results for $\text{BNN}_2$ : $[784, 500, 500, 10], w_s = 19.07\%$. Remarkable improvements can be observed for images $20, 28, 29$ and $33$.

| Image Index | $\tau^1_{\text{tighter,cs}}$ | | $\tau_{\text{Soft-MILP}}$ | |
|:---:|:---:|:---:|:---:|:---:|
| | bound | $t\,(s)$ | bound | $t\,(s)$ |
| $\delta_{\|.\|_\infty} = 0.75$ | | | | |
| 11 | 19.54 | 42.31 | infeasible | 13.39 |
| 20 | 8.72 | 107.36 | timeout | $> 600$ |
| 26 | 14.90 | 37.25 | infeasible | 10.61 |
| 28 | 14.26 | 75.15 | infeasible | 227.07 |
| 29 | 7.48 | 47.33 | infeasible | 200.02 |
| 33 | 7.31 | 49.35 | timeout | $> 600$ |
| 55 | 25.14 | 28.96 | infeasible | 4.67 |
| 72 | 35.33 | 35.80 | infeasible | 6.85 |
| 83 | 22.14 | 44.61 | infeasible | 16.26 |

Table 6: Verification against $\|.\|_2$-attacks: detailed numerical results for $\text{BNN}_1$ : $[784, 500, 500, 10]$, $w_s = 34.34\%$. Notice that the number of timeouts is significant. On the other hand, the verification time associated to $\tau^1_{\text{tighter,cs}}$ remains relatively stable.

| Image Index | $\tau^1_{\text{tighter,cs}}$ | | $\tau_{\text{Soft-MINP}}$ | |
|---|---|---|---|---|
| | bound | $t\,(s)$ | bound | $t\,(s)$ |
| $\delta_{\|.\|_2} = 20$ | | | | |
| 4 | 4.20 | 12.79 | timeout | $> 600$ |
| 10 | 0.71 | 16.41 | infeasible | 564.03 |
| 11 | 6.05 | 13.91 | infeasible | 376.86 |
| 12 | 14.34 | 8.00 | infeasible | 181.41 |
| 13 | 5.03 | 21.96 | timeout | $> 600$ |
| 14 | 34.12 | 15.05 | infeasible | 47.93 |
| 15 | 4.35 | 18.58 | timeout | $> 600$ |
| 23 | 9.91 | 9.13 | infeasible | 0.40 |
| 24 | 37.52 | 12.37 | infeasible | 354.42 |
| 26 | 32.32 | 11.12 | infeasible | 0.71 |
| 28 | 58.43 | 22.14 | infeasible | 203.53 |
| 31 | 27.73 | 8.31 | infeasible | 1.05 |
| 48 | 12.48 | 16.48 | timeout | $> 600$ |
| 49 | 14.85 | 15.72 | timeout | $> 600$ |
| 52 | 34.45 | 10.35 | infeasible | 0.07 |
| 55 | 42.34 | 5.32 | infeasible | 0.08 |
| 56 | 15.68 | 11.15 | timeout | $> 600$ |
| 57 | 30.32 | 6.16 | infeasible | 0.48 |
| 59 | 17.66 | 27.45 | timeout | $> 600$ |
| 61 | 8.08 | 15.45 | infeasible | 102.50 |
| 65 | 5.20 | 8.99 | timeout | $> 600$ |
| 69 | 12.73 | 19.38 | infeasible | 272.38 |
| 70 | 6.37 | 15.82 | infeasible | 292.86 |
| 72 | 46.78 | 15.88 | infeasible | 0.15 |
| 73 | 18.78 | 11.73 | infeasible | 14.15 |
| 77 | 22.70 | 30.85 | infeasible | 480.90 |
| 80 | 11.89 | 12.12 | infeasible | 72.41 |
| 83 | 43.12 | 20.10 | infeasible | 60.96 |
| 86 | 9.89 | 14.54 | infeasible | 531.24 |
| 89 | 18.74 | 17.25 | infeasible | 229.98 |
| 91 | 11.59 | 18.89 | infeasible | 505.02 |
| 92 | 13.55 | 34.95 | timeout | $> 600$ |
| 94 | 7.13 | 14.75 | infeasible | 62.56 |
| 98 | 2.64 | 13.02 | infeasible | 165.10 |
| 99 | 13.27 | 12.16 | infeasible | 100.43 |
| 100 | 42.32 | 8.95 | infeasible | 0.09 |
| $\delta_{\|.\|_2} = 30$ | | | | |
| 14 | 11.81 | 17.54 | timeout | $> 600$ |
| 24 | 15.04 | 28.94 | timeout | $> 600$ |
| 26 | 15.48 | 7.28 | infeasible | 108.11 |
| 28 | 35.08 | 40.25 | timeout | $> 600$ |
| 31 | 6.89 | 16.39 | timeout | $> 600$ |
| 52 | 16.78 | 10.86 | infeasible | 191.03 |
| 55 | 21.80 | 10.49 | infeasible | 555.08 |
| 57 | 12.04 | 10.93 | timeout | $> 600$ |
| 72 | 35.81 | 12.45 | infeasible | 353.96 |
| 73 | 3.59 | 10.44 | timeout | $> 600$ |
| 77 | 3.45 | 37.65 | timeout | $> 600$ |
| 83 | 26.76 | 22.78 | timeout | $> 600$ |
| 100 | 25.51 | 9.14 | infeasible | 64.95 |

Table 7: Verification against $\|.\|_2$-attacks: detailed numerical results for $\text{BNN}_2$ : $[784, 500, 500, 10], w_s = 19.07\%$. Images 57 and 89 demonstrate that $\tau^1_{\text{tighter,cs}}$ can typically certify robustness more than times faster.

| Image Index | $\tau^1_{\text{tighter,cs}}$ | | $\tau_{\text{Soft-MINP}}$ | |
|:---:|:---:|:---:|:---:|:---:|
| | bound | $t\,(s)$ | bound | $t\,(s)$ |
| $\delta_{\|.\|_2} = 15$ | | | | |
| 2 | 2.74 | 31.34 | infeasible | 279.66 |
| 4 | 0.14 | 47.94 | timeout | $> 600$ |
| 11 | 50.95 | 17.32 | infeasible | 12.99 |
| 14 | 19.98 | 39.04 | timeout | $> 600$ |
| 20 | 33.38 | 88.27 | timeout | $> 600$ |
| 23 | 13.43 | 19.91 | infeasible | 253.64 |
| 26 | 40.90 | 16.40 | infeasible | 12.57 |
| 28 | 40.87 | 45.39 | infeasible | 73.84 |
| 29 | 39.00 | 23.42 | infeasible | 9.52 |
| 31 | 15.00 | 15.89 | infeasible | 59.66 |
| 33 | 30.05 | 51.76 | infeasible | 134.40 |
| 48 | 16.31 | 47.78 | timeout | $> 600$ |
| 55 | 53.65 | 12.11 | infeasible | 0.25 |
| 57 | 6.16 | 25.38 | timeout | $> 600$ |
| 61 | 2.29 | 19.44 | infeasible | 253.44 |
| 69 | 20.93 | 67.48 | timeout | $> 600$ |
| 71 | 18.96 | 17.43 | infeasible | 88.33 |
| 72 | 61.78 | 14.47 | infeasible | 1.40 |
| 73 | 25.72 | 14.60 | infeasible | 62.32 |
| 77 | 2.48 | 37.19 | infeasible | 510.44 |
| 83 | 47.72 | 23.62 | infeasible | 81.36 |
| 86 | 6.54 | 115.93 | timeout | $> 600$ |
| 87 | 22.33 | 69.43 | infeasible | 344.06 |
| 88 | 15.48 | 23.39 | infeasible | 250.97 |
| 89 | 27.29 | 29.73 | timeout | $> 600$ |
| 92 | 23.51 | 57.17 | timeout | $> 600$ |
| 100 | 15.40 | 20.03 | infeasible | 108.08 |

**Experimental results for more complex data sets and larger networks:** To further demonstrate the versatility of our method, we provide additional illustrative experiments for $\text{BNN}_3$ : $[3072, 5000, 800, 10], w_s = 55.97\%$, achieving the test accuracy of $47.66\%$ on CIFAR-10 data set.

As illustrated in Table 8, our approach demonstrates comparable performance even on larger datasets, such as CIFAR-10, and for larger networks involving nearly 9000 neurons.

Specifically, $\tau^1_{\text{tighter,cs}}$ proves the robustness of images 1, 2 and 35 at least 3x, 6x, and 5.5x faster than $\tau_{\text{Soft-MILP}}$, respectively. In contrast, the low quality of LP bounds prevents MILP from providing an answer within the imposed one-hour time limit. These additional experimental results further validate that our method consistently provides high-quality lower bounds, even for large-scale problems, which is concordant with the results for $\text{BNN}_1$ and $\text{BNN}_2$.

However, for images indexed by 8 and 10, for example, our method is unable to provide an answer, while MILP can efficiently certify their *non-robustness*. This difference arises due to the fact that MILP based methods do not aim to solve the optimization problems to global optimality but instead focus on determining the feasibility of an adversarial attack, which is inherently less computationally demanding.

We believe that incorporating our tighter SDP bounds within the MILP framework could enhance the ability of MILP methods to certify *robustness* in more complex cases. This represents a promising direction for future research.

Table 8: Verifying robustness of $BNN_3$ on CIFAR-10, for an input region determined by $\delta_{||.||_\infty} = 0.2/255$. Data were scaled to $[-1, 1]^{3072}$, and a time limitation of $3600\,s$ was imposed. We present the results of the robustness verification queries on the first $40$ images from the test data set, among which $22$ were correctly classified.

| Image Index | $\tau^1_{\text{tighter,cs}}$ bound | $t\,(s)$ | $\tau_{\text{Soft-MILP}}$ bound | $t\,(s)$ |
|---|---|---|---|---|
| 1 | 20.46 (robust) | **1179.55** | timeout | $> 3600$ |
| 2 | 7.83 (robust) | **571.08** | timeout | $> 3600$ |
| 7 | -30.28 (unknown) | 380.10 | timeout | $> 3600$ |
| 8 | -68.50 (unknown) | 1721.79 | feasible (not robust) | 1.71 |
| 10 | -62.31 (unknown) | 1330.03 | feasible (not robust) | 87.46 |
| 11 | -142.50 (unknown) | 650.79 | feasible (not robust) | 0.11 |
| 12 | -94.92 (unknown) | 1115.18 | feasible (not robust) | 0.071 |
| 14 | -103.36 (unknown) | 493.80 | feasible (not robust) | 0.067 |
| 15 | -71.09 (unknown) | 2090.44 | feasible (not robust) | 57.09 |
| 18 | -97.96 (unknown) | 1469.67 | feasible (not robust) | 0.08 |
| 19 | 31.06 (robust) | 344.70 | infeasible (robust) | 9.02 |
| 20 | -25.66 (unknown) | 823.54 | timeout | $> 3600$ |
| 21 | -90.54 (unknown) | 670.94 | feasible (not robust) | 0.59 |
| 24 | -29.08 (unknown) | 629.50 | timeout | $> 3600$ |
| 27 | -83.76 (unknown) | 766.95 | feasible (not robust) | 0.54 |
| 29 | -110.26 (unknown) | 873.51 | feasible (not robust) | 0.07 |
| 30 | -60.63 (unknown) | 1407.68 | feasible (not robust) | 1.39 |
| 31 | -29.51 (unknown) | 1190.56 | timeout | $> 3600$ |
| 33 | timeout | $> 3600$ | timeout | $> 3600$ |
| 34 | -73.15 (unknown) | 569.46 | feasible (not robust) | 0.51 |
| 35 | 36.62 (robust) | **657.73** | timeout | $> 3600$ |
| 40 | -52.82 (unknown) | 650.79 | timeout | $> 3600$ |

