# OpenReview forum: "Verifying Properties of Binary Neural Networks Using Sparse Polynomial Optimization"
_ICLR.cc/2025/Conference — ICLR 2025 Poster_

### Official Review · Reviewer_o79c · 2024-10-24

**Soundness:** 2
**Presentation:** 2
**Contribution:** 3
**Rating:** 6
**Confidence:** 3

**Summary:**

The paper considers the formal verification of binary neural networks in the classification setting.
Formal verification is necessary as (any) neural network is inherently prone to adversarial attacks.
In the paper, the robustness of binary neural networks is verified using semidefinite programming derived from polynomial optimizations.
In particular, polynomials up to order 2 are used as constraints and existing solvers (Mosek, Gurobi) are used to prove the robustness property by providing a lower bound.
The experients considers l_2 and l_inf robustness properties and shows that while achieving better bounds than simple linear programming, it also has a reduced verification time than related methods based on MILP/MINP.

**Strengths:**

- the considered problem is highly relevant
- deep theoretical analysis that goes beyond simple linear relaxations and rather design the verification algorithm to exploit certain properties of the considered network architecture
- The theoretical results show strictly better bounds (Thm. 3.1 & 4.1), which proves the point of using the more complex verification algorithm.
- the approach allows to verify more general input perturbations. In particular, l_2, which was not considered before in binary neural network verification although it has been considered in standard neural networks.

**Weaknesses:**

Main points:
It is hard to follow the theoretical analysis in the paper. I tried to identify some of the key issues and give examples for each case:
- Crucial results from references are used without re-stating them or at least providing the exact equation number in the reference paper:  E.g., Sec. 4: "Firstly, notice that the semi-algebraic representation of the subgradient of the ReLU function derived in Chen et al. (2020) provides an alternative encoding of the sign(·) function." This sentence in then used without further explanation to transform (7) into (17).
- Formal inconsistencies: E.g., it is unclear what the output of sign(0) is; (4) says that W can be in {-1,0,1} but later in Sec. 3 says it is in {-1,1}, which might influence how certain normalizations have to be considered, e.g., the bounds in (11).
- try to introduce all concepts before they are used. E.g., the term "cliques" is not properly introduced. It appears to be sets of neurons but I am unable to assess how they are determined. Also, the variable "d" is used throughout Sec. 2.3 but is only later introduced as "relaxation order". Similarly, the operation "nv(A)" is introduced after its first usage in (12), making (12) unable to be understood by the reader up to this point. It is explained in the paragraph after, but I think moving such things to the notation section would be more beneficial as they are used throughout the paper and a reader can then go back to the notation section to (re-)refer to those details.
- Steps to derive the constraints are rather quick and more explanations could help to follow along

Additionally, the experiments show only single-dimensional results without saying over how many instances the results are averaged and do not provide a standard deviation where applicable. Also, it is unclear if the compared approaches are taken from the literature or arbitrarily constructed. If the latter is true, it misses a comparison to related work altogether.

Minor points:
- It is unclear why the approach does not suffer from floating point inaccuracies. In fact, the term "floating-point" only appears twice in the paper (only in the abstract and the contribution section (Sec. 1.1)).
- the paper claims that the approach does not suffers from an exponential running time (Sec. 6) but it misses a thorough analysis of its runtime. The average speed up of 4.5 / 11.4 stated in the contribution section (Sec. 1.1) does also not support this claim.
- Make the example in Fig. 1 a running example by moving it further up in the paper and continuously refer to it when explaining all terms.
- Similarily, give an intuitive explanation why adding those tautologies are necessary
- Only the classification setting is considered. This could be stated more clearly.
- the related works section could also include more research on the verification of standard neural networks (e.g., VNN-COMP) to better place this line of research in the broader context.
- Fig 2: misses a label and ticks of the x axis.
- no repeatability package is provided

Further points that could help improve the paper but did not directly influence the score:
- give a real-life example where binary neural networks are used and verifying the robustness is necessary
- Avoid the usage of abbreviations. These usually do not save a lot of space but makes reading more difficult as one is challenged to memorize all abbreviations in addition to the complexity of the paper.

**Questions:**

- Can you provide the missing details about the evaluation I mentioned above?
- You mentioned in Sec. 2.3 that you use the (tractable) inner approximations of the set of polynomials that are nonnegative on S. Why is it enough to only consider that subset?
- Why can the weights take values {-1,0,1} but the bias any value in |R?
- Why does you appraoch not suffer from floating-point inaccuracies?

---

> ### Author Response · Authors · 2024-11-23
> **Detailed Response to Reviewer o79c**
>
> We are grateful to the reviewer for the nice summary and for acknowledging the strengths of our work. Below, we address all the raised weaknesses and questions in order in which they appear.
>
> ### **Weaknesses:**
>
> **Main points:**
>
> - Our approach is designed for optimization problems where the objective function and constraint set can be encoded semi-algebraically (as intersections of polynomial equalities and inequalities). This is why we specifically focus on encoding the $\text{sign}(\cdot)$ function in this manner, as different encodings yield different levels of bound tightness. For example, the encoding presented in (7) might seem the most standard. However, in (17), we adopt an *alternative* encoding introduced in Chen et al. (2020), since the subgradient of the ReLU function is the $\text{sign}(\cdot)$ function itself. To clarify, (7) was not *transformed* into (17); instead, (17) provides an equivalent but more efficient encoding of (7).
>
>   We would be happy to address any other reference-related confusions in order to enhance the clarity of our work.
>
> - Our encoding method allows $\operatorname{sign}(0)$ to be either $+1$ or $+1$, which can be inferred from (6) and (7). Moreover, the operator $\operatorname{nv}(\cdot)$ defined after (12b) is used to handle the general cases (normalizations) where $w \in${-1, 0, 1}. If
>  $w \in${-1, 1}, we always specify it before stating mathematical expressions. The assumptions in Sec. 3 that $w$ cannot be 0 are  mainly for the purpose of facilitating statements and proofs, they do not influence the validity of our main results.  In fact, in our main theorem (Theorem 4.1), we allow the elements of $W^{[1]}$ to be in  $[-1,1]$, and the elements of other weight matrices  to be in {-1,0,1}. We will standardize this
> notation in the main text.
>
> - It is true that *cliques* should be understood as subsets of neurons (decision variables in (19a)-(19d)), and their determination is dictated by the problem structure. More precisely, cliques correspond to subsets of {$1,\dots,n$} such that the two properties stated on Lines 259 and 260 hold. We can replace the term *cliques* by *subsets of variables* in subsequent versions of our work.\
> The variable $d$ is also defined on Line 229, saying "...defines a
> hierarchy of dense SDP relaxations whose size increases with $d$...". We could transform this into "...defines a hierarchy of dense SDP relaxations whose size increases with *relaxation order* $d$...", in order to increase the clarity. \
> We will add the definition of the operator $\text{nv}(\cdot)$ to the notation section.
>
> - We have made efforts to reduce the theoretical complexity of the paper as much as possible. The two main types of constraints were introduced as follows: the network-structure-induced constraints in (7) and the input region constraints on Line 195. These were first combined in (8). All other constraints used in the paper are variations (rather than transformations) of these two types, resulting in different optimization problems with varying levels of bound quality. If the reviewer has specific suggestions on improving the logical flow, we would be happy to incorporate them.
>
> On Line 408, we stated: "We assess the performance of our method (number of solved cases (cert.) and verification time $t(s)$) in verifying robustness of **the first 100 test instances**". Due to the high computational cost of these experiments, re-executing them to derive statistical measures such as standard deviations would be impractical and with little value, as the robustness answers would remain unchanged. \
> As highlighted in our response to Reviewer Rj7X, we compare our method to other optimization-based techniques from the literature, specifically MILP. Other methods are either not optimization-based, incompatible with continuous input spaces and $||.||_2$ attacks, or exhibit lower scalability compared to the two methods evaluated in our work. We aimed to focus on approaches most relevant to the context of our study.
>
> **Minor points:**
>
> - Since SDP solvers are implemented using the floating-point arithmetic, there may be some error polynomial $e$ with extremely small coefficients, and such that the actual output of the solver is $f-\lambda-
> \sigma +e \in Q(g)+I(h)$ instead of  $f-\lambda-\sigma  \in Q(g)+I(h)$.  Since $x  [-1,1]^n$, it is possible to derive a valid lower bound of the form $f \geq \lambda-e^{*}$,
>
>   for some small $e^{*}$  that can be computed from the coefficients of $e$.
>
>   For more details regarding the accuracy of SDP solving procedures, we refer the reader our reference (Magron et al., 2015). To the
>   best of our knowledge, no such guarantees are available for the MILP/MINP methods.

---

> ### Author Response · Authors · 2024-11-23
> **Detailed Response to Reviewer o79c - Continuation 1**
>
> - We would like to point out that our relaxation-based method
> yields optimization problems with number of constraints being polynomial in the number of (clique)
> variables, which gives us confidence that for larger-scale problems, it would demonstrate significantly
> better efficiency compared to existing algorithms such as exact SAT/SMT-based methods, or even MILP, which we found to be up to 50x slower than our method in certifying robustness against some severe cases. In fact, due to the computational complexity of MILP, there is no available polynomial-time algorithm for solving MILP problems (unless P=NP). However, the SDP relaxation problem is essentially a *convex* optimization problem with polynomial   number of  variables and constraints.
>
> - Figure 1 is already closely tied to our running example, *Example 2.1*, where we present the SDP optimization problem derived from a simple network structure with 2 hidden layers, each containing 2 nodes. Figure 1 revisits *Example 2.1* while introducing a key additional concept: correlative sparsity. Finally, in (20), immediately following Figure 1, we integrate sparsity, constraint encoding, and tautologies to provide a concrete example of the optimization problems our method addresses.
>
> - Although the redundant constraints do not appear useful from the primal perspective, as they do not influence the feasible set, we have proven (see Corollary A.3.1) that their
> presence results in a larger quadratic module from the dual perspective. This larger quadratic
> module encapsulates more polynomials and enables better SOS decompositions, which in turn
> results in improved relaxation bounds.
>
> - We refer the reviewer to Line 151, where we stated: "Let $L \geq$ be the number of hidden layers of a *classifying* BNN...". Moreover, Remark 2.1 focuses on adversarial attacks which specifically concern label-altering in classification tasks.
>
> - We aimed to make our related works section as comprehensive as possible, particularly with regard to BNNs. Some methods for standard neural network verification were already included, especially those relying on Semidefinite Programming techniques (see the paragraph on SDP-based verification methods). However, as suggested, we will incorporate additional relevant references to better situate our work within the broader context of neural network verification research.
>
> - In Figure 2, the explanation for x-axis is already given on Line 433: "Each subplot x-axis
> represents *image indices* sorted in the descending order of $\tau_{tighter, cs}^{1}$ values". That is why the red line appears smoother than the other two lines. Also, since those indices would appear in different order for each sub-plot, we believe it would make the interpretation even more difficult while occupying more space.
>
> - We included a zip file in our submission containing the data used in our experiments, along with detailed instructions on how to execute the relevant code which can be found in *readme.md*. Additionally, we will provide an explicit link to the GitHub repository in the non-anonymous version of our paper.
>
> ----
>
> - We thank the reviewer for the suggestion. Indeed, BNNs are widely used in edge devices for tasks such as object detection, image recognition, and decision-making in *resource-constrained environments*, like autonomous delivery drones, for example. Such drones might encounter adversarial environmental conditions, such as lighting variations, or weather disturbances, and failing  to robustly classify objects under these altered environmental conditions could lead to a collision, endangering people or property. We will include these illustrations in the refined version of our work.
>
> - We thank the reviewer for the suggestion. We will try to minimize the use of abbreviations, while adhering to the page number limitations. However, some of the abbreviations that we employed frequently, like SDP, MILP, or BNN, are widely used as autonomous words, and we believe that indicating their full meaning could eventually make the paper more cumbersome.

---

> > ### Author Response · Authors · 2024-11-23
> > **Detailed Response to Reviewer o79c - Continuation 2**
> >
> > ### **Questions:**
> >
> > - We refer the reviewer to our previous responses, where we have provided detailed explanations addressing all the points raised. If there are any specific aspects still unclear, we would be happy to provide additional clarification.
> >
> > -  A dual reformulation of the polynomial optimization problems requires verifying a difficult positivity constraint of a general polynomial, which is known to be intractable. However, requiring the polynomial to be a sum of squares of other polynomials is tractable, as it can be done using Semidefinite Programming. Since the set of polynomials that are sum of squares represents a proper subset of nonnegative polynomials, this tractable positivity constraints induce a valid convex relaxation of the initial problem. Moreover, by increasing the relaxation order, and under some technical conditions, we can expect the lower bound from the relaxation to converge to the true optimal value, generally in finite amount of steps. This is the case because the subset of sum of squares polynomials is dense in the set of positive polynomials.
> >
> > - Requiring the weights in BNNs to be in {$-1,0,1$} is the cornerstone of their computational efficiency. However, imposing the same constraint on the bias parameter $\textbf{b}_j^{[i]}$ would make the bias almost irrelevant, as $\textbf{b}_j^{[i]}$ would be able to influence the output of a node $\textbf{x}^i_j$
> >
> >   only if $\sum_{j=1}^{n_{i-1}} \textbf{x}^{i-1}_j w^i_j =0$,
> >
> >
> >   where $w^i_j=\textbf{W}^{[i]}_{(j,:)}$.
> >
> > - We refer the reviewer to our response to the first question in the **Minor points** section.

---

> > > ### Comment · Reviewer_o79c · 2024-11-26
> > >
> > > Thank you for your detailed answer. I raised my score as some of the points were addressed, but I still think it would improve the paper by a lot if my points and also points raised by others are adequately incorporated into the paper. This includes improvements in (i) notation, (ii) visualization, and (iii) experiments.
> > >
> > > Apart from my comments above, I give examples for each of these points here while re-reading the paper:
> > > - (i) Notation: Line 128-129: The k-th row is indexed by $A_{(k\colon,)}$ (did you mean $A_{(k,\colon)}$?) Why is the i,j-th entry then $A_{i,j}$ instead of $A_{(i,j)}$? Also, I find $x^2$ confusing as it usually means "x squared" instead of a second vector Maybe use $x_2$? The k-th entry would then be $x_{2(k)}$ (also not optimal, but consistent with the notation above).
> > > - (ii) Visualization: I would really like to see a running example showcasing the computed bounds and relaxations in the two-dimensional output space and not just the cliques as in Fig. 1. I think this would really strengthen the otherwise rather dry Sec. 3 and 4. You might also want to look at how such visualizations were done in related work with non-trivial relaxation in standard NNV [1,2].
> > > - (iii) Regarding question 1 of reviewer Rj7X: You mentioned that your approach might scale better than the others (although the quality of the bounds might suffer). Show it! This would strengthen your claim by a lot.
> > >
> > > Overall, I find the direction very promising. Best of luck with your submission!
> > >
> > > [1] Fatnassi, W., et al. "BERN-NN-IBF: Enhancing Neural Network Bound Propagation Through Implicit Bernstein Form and Optimized Tensor Operations." IEEE Transactions on Computer-Aided Design of Integrated Circuits and Systems 43.11 (2024): 4334-4345.
> > >
> > > [2] Ladner, T., and Althoff, M. "Automatic abstraction refinement in neural network verification using sensitivity analysis." Proceedings of the 26th ACM International Conference on Hybrid Systems: Computation and Control. 2023.

---

> > > > ### Author Response · Authors · 2024-11-30
> > > > **Detailed Response to Reviewer o79c - Continuation 3**
> > > >
> > > > We are very grateful to the reviewer for acknowledging the full potential of our work and for providing the additional feedback.
> > > >
> > > > (i) We thank the reviewer for the suggestions regarding the presentation aspect of our paper. We have tried to modify some notations to avoid ambiguity, notably concerning matrices and vectors. We have also included more precise references regarding the theoretical results upon which our approach is dependent. Some examples regarding the relevance of the considered problem were also provided in the introduction. We invite the the reviewer to consult these, and other modifications, in the revised version of our paper (changes are highlighted in blue).
> > > >
> > > > (ii) We thank the reviewer for providing these additional references. However, creating similar illustrations, even for our toy BNN example would be very challenging. The relaxations proposed in [1] and [2] do not involve any *lifting*, meaning that the feasible set remains within the same decision space as the original problem. In contrast, SDP relaxations involve introducing new variables, representing the moments of some unknown measure, and requiring these new variables to satisfy specific semidefinite positivity constraints. Consequently, the new feasible set is a spectrahedron in this higher-dimensional space. If one would try to project this set into the original space, one would end up with the convex hull of the graph of the $\operatorname{sign}(\cdot)$ function, and thus unable to perceive any difference with respect to LP relaxations.
> > > > That is why we opted for analytical proofs of the superiority of our relaxations, as stated in Theorem 4.1, and demonstrated experimentally in Table 3, for example.
> > > >
> > > > Finally, our running example is meant to illustrate the nature of the problems we solve. Example 2.1, Figure 1, and equation (20) allow the reader to understand how the network structure determines the sparsity pattern of the problem, and how this sparsity pattern in turn translates into matrices (decision variables) of smaller size, highlighting the efficiency of the proposed approach.
> > > >
> > > > (iii) Regarding performance evaluations of our method, throughout the paper, this was something that has been repeatedly highlighted in many ways. Specifically, *Theorem 4.1*, *Table 3*, and the additional *Table 8* demonstrate, **both theoretically and experimentally**, that our bounds consistently outperform LP bounds, even in deeper networks (up to 6 layers) with a large number of hidden nodes (reaching as many as 9000).
> > > >
> > > > We emphasize once again that no other method in the literature, capable of simultaneously addressing continuous input spaces and $||.||_2$ attacks, has been able to achieve this level of scalability.
> > > >
> > > > Notably, additional experiments conducted at the request of reviewers Rj7X and TkGW reveal that the MILP method, which relies on LP bounds, is only capable of proving the robustness of a single image within the 1h time limit.
> > > >
> > > > Thus, MILP method can be very useful for finding attacks (since the attack feasibility problem is much easier), but they cannot be expected to provide robustness certificates more often than our method. Similar reasoning applies for other SAT/SMT-based, exact methods, if one would try to make them compatible with continuous input spaces and $||.||_2$ attacks (more details on this can be found in our latest reply to the reviewer 7uz5).

---

> > > > > ### Comment · Reviewer_o79c · 2024-12-02
> > > > >
> > > > > Thank you for your answer and for clarifying the notation, the description of the visualization, and for addressing my questions about the experiments. I raised my score because of it.

---

### Official Review · Reviewer_7uz5 · 2024-11-03

**Soundness:** 3
**Presentation:** 3
**Contribution:** 2
**Rating:** 5
**Confidence:** 4

**Summary:**

This paper introduces a novel approach to verify the robustness of BNN based on sparse polynomial optimization, specifically through Semidefinite programming relaxation. The authors first encode the original verification problem as a polynomial optimization problem (POP) and then apply SDP relaxation on it to obtain lower bounds for certifying the robustness. While the verification method is sound and incomplete, experimental results show that it provides a more precise bound than LP-based methods.

**Strengths:**

- The problem (verification of Transformers) is important.
- A novel method based on SDP to verify the robustness of BNN.
- Compared to LP-based methods, a more precise lower bound is obtained.

**Weaknesses:**

- The motivation for this approach is not intuitive. The authors claim that existing methods are either incompatible with continuous input data or are limited to $L_\infty$ input perturbations. However, expanding support to other types of input regions may not constitute a compelling contribution: i) $L_\infty$ perturbations are the primary standard in the field, and ii) most BNN verifiers support continuous input spaces. Furthermore, for methods $M$ specifically designed for discrete input region (i.e., input $x\in$ {-1,1}$^{n_0}$), one can also treat the second layer (assuming the original input region is continuous, as the setting in this work) as the new input layer and then apply these methods $M$ to verify the robustness.
- The experimental results are not convincing. The MILP-based method, which is sound and complete, solves significantly more verification tasks than the proposed SDP-based approach. While the authors introduce a "soft" encoding to avoid numerical errors, it is relatively straightforward similar to methods in previous work.
- For $L_\infty$ experiments, it would be beneficial to include comparisons with other SOTA methods, such as the SMT-based approach (Amir et al.).
- As Gurobi supports multi-threaded solving, presenting experimental results on multiple threads would also strengthen the evaluation.
- Table 3 gives more results on bound computation, however, it is not clear if these bounds lead to verified results (i.e., proving the robustness).

**Questions:**

See the weakness raised in **Weaknesses**.

Other minor comments:

- I failed to find the explanation for what the data in parentheses in Column $\tau_{tighter,cs}^1$ represent in Tables 1 and 2.
- MILP methods aim to verify whether a property holds or fails, offering a definitive answer rather than estimating bounds to certify robustness. Therefore, comparing the two approaches (SDP-based vs. MILP-based) should ideally focus on metrics related to verification success rates, computational efficiency, or scalability across various network sizes, rather than on bound tightness.

---

> ### Author Response · Authors · 2024-11-23
> **Detailed Response to Reviewer 7uz5**
>
> We are grateful to the reviewer for the nice summary and for acknowledging the strengths of our work. Below, we address all the raised weaknesses and questions in order in which they appear.
>
> ### **Weaknesses:**
>
> - We acknowledge that most works in the domain of BNN verification primarily focus on $||.||_\infty$ perturbations. However, this should not overshadow the importance of addressing $||.||_2$ attacks. In fact, a significant body of impactful research has been conducted on $||.||_2$ robustness verification for full-precision (non-quantized) neural networks, including the works in [1, 2, 3] mentioned below.
>
>   While $||.||_\infty$ robustness emphasizes localized, worst-case scenarios, $||.||_2$ robustness remains essential for understanding a
>   model’s behaviour under more natural, distributed perturbations. These two metrics are complementary, and studying both provides a
>   more comprehensive evaluation of a model's reliability.  That said, current BNN verification methods appear incapable of addressing
>   the additional complexity introduced by the non-linearities inherent in $||.||_2$ attacks.
>
>   Moreover, we argue that there is a fundamental difference between continuous and discrete input spaces when it comes to robustness
>   verification. As highlighted in (Ivashchenko et al., 2023), input quantization is an artificial step designed to improve the efficiency of
>   the verification process. Since networks are generally trained to operate on continuous input spaces, quantization can reduce
>   network's post-training accuracy. Verifying BNNs on continuous input spaces allows for the consideration of all possible input states,
>   thereby enhancing the certainty of verification. However, when applied to continuous input spaces, current BNN verification methods
>   face two major challenges: scalability and the inability to provide reliable bounds. Continuous input spaces induce numerous
>   numerical errors and lead to ill-conditioned optimization problems, as noted in our paper on Line 438. Thus, adaptations proposed by
>   the reviewer,  although possible, are not always straightforward and may lead to inefficiencies or loss of precision in verification.
>
>   For instance, one of the difficulties is providing an exact description of the input region for method *M*. This requires an oracle capable
>   of precisely characterizing the image set of a single-layer BNN with continuous input space. Specifically, it must describe all elements
>   of the set  {$y \in $ {$-1,1$ }$^{n_1} | \exists x \in \mathcal{B} \subseteq [-1,1]^{n_0}, y = \operatorname{sign}(Wx+b)$}, given a region
>   $\mathcal{B}$ and some parameters $(W,b)$. This problem can be reformulated as a maximum feasible subsystem problem, which is a
>   well-known *NP-hard* problem, by optimizing a linear objective function over this set.
>
>   Finally, let us provide an illustrative example showcasing the difficulty of handling continuous input spaces and the necessity of
>   considering floating point errors:
>
>   Consider a BNN with input $\mathbf{x}:=(x_1,x_2)\in[0,255]^2$ and one hidden layer $\mathbf{y} :=(y_1,y_2,y_3)$, where:
>
>   $y_1=\text{sign}(x_1+127.5)$,
>
>   $y_2=\text{sign}(x_2+127.5)$,
>
>   $y_3=\text{sign}(255+\varepsilon-x_1-x_2)$,
>
>   with  $\varepsilon>0$. Let
>   the objective function be $f:(\mathbf{x},\mathbf{y}) \mapsto y_1+y_2+y_3+1.5$.
>
>   If the input space of this BNN is discrete, meaning
>   $x_1$ and $x_2$  are integer-valued, then using the encoding method from (Narodytska et al, 2018), would lead to the following
>   integer linear programming formulation:
>
>
>   $\min_{\mathbf{x},\mathbf{y}} f(y_1,y_2,y_3)=y_1+y_2+y_3+1.5$
>   s.t.
>     $x_1 < C_1  \implies y_1=-1, x_1 \geq C_1  \implies y_1=1$,
>
>     $x_2< C_2   \implies y_2=-1, x_2 \geq C_2   \implies y_2=1$,
>
>     $-x_1-x_2< C_3   \implies y_3=-1, -x_1-x_2\geq C_3   \implies y_3=1,$
>
>   where $C_i$ refer to the values obtained by rounding the some constants up to the nearest integer.
>
>   Specifically,  $C_1=C_2=128$ and $C_3=\lceil-  255-\varepsilon \rceil $. Thus, as long as $0<\varepsilon<1$, problem will remain
>   unchanged and will always give the true solution  $f^{*}=0.5$.
>
>   However, for a continuous input space,  when  $\varepsilon$ is very
>   close to 0 (in practice when $\varepsilon <10^{-6}$), the MILP solver may output a spurious result, say$f^{*}=-1.5$.
>
>     -  [1] Jeremy Cohen, Elan Rosenfeld, Zico Kolter. "Certified Adversarial Robustness via Randomized Smoothing". Proceedings of the
>        36th International Conference on Machine Learning, PMLR 97:1310-1320, 2019.
>     - [2] Sahil Singla, Surbhi Singla, Soheil Feizi. "Improved deterministic $||.||_2$ robustness on CIFAR-10 and CIFAR-100".
>       International Conference on Learning Representations, 2022.
>     - [3] Xiaojun Xu, Linyi Li, Bo Li. "LOT: Layer-wise Orthogonal Training on Improving $||.||_2$ Certified Robustness". Conference on
>       Neural Information Processing Systems, 2022.

---

> ### Author Response · Authors · 2024-11-23
> **Detailed Response to Reviewer 7uz5 - Continuation**
>
> - MILP/MINP methods being exact, it does allow them to prove both robustness and non-robustness. Consequently, it is not surprising that they solve more verification tasks within a sufficiently large time period. However, as shown in Table 1 and Table 2, the gap in the number of solved instances significantly reduces under more severe or $||.||_2$ attacks, where the strengths of our method become more apparent.\
> It is also important to note that we did not require branch-and-bound-based MILP/MINP methods to compute *exact* optimal values of the optimization problems. Doing so would have led to significantly more timeouts, rendering this approach impractical for larger or more complex networks. Instead, we allowed the MILP/MINP solvers to terminate as soon as a nonnegative bound was obtained, while our SDP solver solved the relaxed optimization problems to optimality.\
> Lastly, the introduction of the Soft-MILP approach was not intended as a claim of novelty. Rather, it was designed to highlight the intrinsic limitations and unreliability of MILP bounds when applied to challenging combinatorial and ill-conditioned problems.
>
> - We thank the reviewer for the suggestion. Our primary focus was to compare our method with other optimization-based BNN verification techniques. Additionally, the network studied in (Amir et al., 2021) consists of only 300  hidden neurons, and it is unlikely that their approach can scale effectively to the larger networks we considered. Moreover, efficient SAT methods, such as those proposed in (Jia \& Rinard, 2020a), rely heavily on input space quantization, restricting their applicability to the framework we considered.
>
> - On Line 397, we described the hardware used for our experiments and would like to clarify that Gurobi was already utilizing its default multi-threading configuration. While we acknowledge that fine-tuning the number of threads could potentially enhance performance, our primary focus was to ensure a fair comparison across different optimization methods, rather than adjusting solver-specific settings.
>
> - Instances presented in Table 3 in Appendix 3 were generated randomly. From our experience, randomly generated instances are unlikely to exhibit robustness. Despite this, we can still use these instances to obtain meaningful information regarding both the quality of the bound and the solving time. Naturally, an algorithm that can compute tighter bounds can  verify more instances as well. This is further demonstrated in Table 1, where our improved SDP bounds always outperform the LP bounds in terms of the number of certifiably robust cases.
>
> ### **Questions:**
>
> - On Line 421 we stated: "The runtime in parentheses refers to
> the average runtime over the instances that our method verified successfully". For instance, when $\delta_{||.||_\infty}=1.50$, our method successfully verifies 6 instances with an average time of  $26.99 s$ , while for the MILP solver, the average run time over those 6 instances is  $191.95 s$.
>
> - We would like to emphasize that the importance of bound tightness should not be understated, as MILP methods are also inherently dependent on bound tightness. Specifically, branch-and-bound based MILP approaches relax the discrete constraints $x_i \in$ {0,1} into linear constraints $x_i \in [0,1]$. In our paper, we demonstrate, both theoretically and experimentally, the limitations of these linear bounds, such as those used by MILP solvers like Gurobi, in the context of BNN robustness verification. For large-scale problems, *tight bounds are critical*, as overly conservative LP relaxations can result in an excessively large number of branching steps.
>
>   One of the key suggestions of our paper, as pointed out by the reviewer Rj7X,  is to explore replacing these LP bounds with
>   tighter SDP bounds. Our experimental results indicate that developing SDP-based branch-and-bound solvers could lead to
>   significant progress when it comes to addressing challenging optimization problems, such as the ones studied here.

---

> > ### Comment · Reviewer_7uz5 · 2024-11-26
> >
> > Thanks for your detailed answers. With some of my concerns clarified, I have adjusted my score. However, I still question the motivation for exploring the continuous input space for BNNs as I really don't think it should pose a technical issue.
> > However, I agree with the authors on the importance of addressing other norm-based attacks after reading the comments. I also concur with the authors and other reviewers that tighter bounds might significantly enhance verification scalability, like when combined with B&B algorithms. Nevertheless, I remain concerned about the effectiveness and efficiency of its "robustness verification" performance (not bound computation results) compared to SOTA tools (or when combined with the B&B algorithm), particularly in the paper's current state.

---

> ### Author Response · Authors · 2024-11-30
> **Detailed Response to Reviewer 7uz5 - Continuation 2**
>
> We thank the reviewer for additional suggestions.
>
>
>
> 1) Regarding the continuous input argument, let us consider the reference (Jia \& Rinard, 2020a) from our paper. This work introduced a novel SAT solver that is currently among the most efficient and scalable BNN verification tools. These authors themselves argue at page 3:
>
> ------
> "*The first layer of a BNN is usually applied on floating point inputs or fixed-point numbers. However, **encoding floating-point or integer arithmetic in SAT typically incurs high complexity**. To **simplify** the verification process, we quantize the inputs:*
> $$ x^q = \left\lfloor \frac{x}{s} \right\rceil \cdot s, $$
> *where $x \in \mathbb{R}^n_{[0,1]}$ is the real-valued input, $x^q$ is the quantized input to be fed into the BNN, and $s$ is the quantization step size which can be set to $s = \frac{1}{255}$ for emulating 8-bit fixed-point values, or $2\epsilon$ for adversarial training with a $\ell_\infty$ perturbation bound of $\epsilon$. Since a robust network should be invariant to perturbations within $[x - \epsilon, x + \epsilon]$, we **expect** the quantization with $s = 2\epsilon$ not to discard information useful for robust classification, which is confirmed by checking that a few choices of the quantization step do not noticeably affect test accuracy.*"
> _______________________________________
>    - As previously discussed, since BNNs are both trained and typically deployed on continuous data, it is natural to expect them
>    to be verified on continuous data as well.  However, robustness verification algorithms that rely on input discretization fail to
>    account for all possible input configurations. In essence, input quantization serves only as a heuristic, as there is no
>    guarantee that the outcomes of robustness queries will remain consistent before and after quantization, as suggested by the
>    last sentence of the cited paragraph.
>
> Moreover, on page 4, the authors develop:
>
> __________________
> "*Recall that inputs are quantized by $x^q = \left\lfloor \frac{x}{s} \right\rceil \cdot s$, which enables us to only encode the integer interval of allowed $\left\lfloor \frac{x}{s} \right\rceil$ values by merging the multiplier $s$ into $k^{\text{BN}}$ of the first layer. Encoding the constraint $v = \left\lfloor \frac{x}{s} \right\rceil \in \mathbb{Z} \cap [a, b]$ is achieved by introducing $b - a$ auxiliary Boolean variables $\{t_1, \cdots, t_{b-a}\}$ and assigning $v = a + \sum_{i=1}^{b-a} t_i$.*"
> ___________________
>    - This further illustrates that input quantization is an artificial step used to significantly enhance the verification procedure, which
>    would otherwise become rather impractical. Indeed, in non-discretized scenarios, the number of Boolean variables $\{t_i\}$
>    would very quickly become *intractable*.
>
> 2) Regarding the *robustness verification performance*, we argue that our approach, supported by the additional results in Table 8, demonstrates clear advantages in delivering reliable bounds that can guarantee robustness, particularly under challenging scenarios such as severe attacks or $||\cdot||_2$-norm perturbations. While MILP-based methods can be effective in finding counterexamples to **prove non-robustness**, they struggle significantly if required to **certify robustness**. Similarly, exact SAT-based methods, though effective in discrete settings, become impractical when addressing continuous input spaces or nonlinear perturbations like
> $||\cdot||_2$-norm attacks.
>
>    We believe that our approach should not be viewed as directly competing with existing methods but rather as complementary,
>    as it  aims to address scenarios that are particularly challenging for traditional techniques, such as proving robustness in
>    continuous domains with complex perturbation models. Finally, we hope that our work has the potential to  encourage the
>    research community to focus on developing more efficient and scalable SDP solvers, which would have many significant
>    repercussions in different domains.

---

### Official Review · Reviewer_Rj7X · 2024-11-04

**Soundness:** 3
**Presentation:** 3
**Contribution:** 3
**Rating:** 6
**Confidence:** 3

**Summary:**

The paper introduces a novel method for verifying the robustness of Binary Neural Networks (BNNs) against adversarial attacks using Semidefinite Programming (SDP) relaxations derived from Sparse Polynomial Optimization. This approach outperforms existing LP relaxation used in MILP-based verification methods without introducing much computation overhead. Specifically, the authors suggest that the SDP-based relaxations could be embedded within branch-and-bound algorithms in MILP solvers to improve bound estimation, accelerating the verification process without altering the core MILP framework. Their method achieves bounds that are up to 55% tighter than those obtained with traditional linear relaxations and demonstrates significant computational efficiency, especially under large input perturbations.

**Strengths:**

1. **Novel Use of SDP for BNN Verification**: Employing sparse Polynomial Optimization and SDP relaxations for BNN verification represents an innovative approach, potentially enhancing scalability and precision over existing MILP-based methods.

2. **Significantly Improved Bounds**: By using SDP relaxations, the authors achieve up to 55% tighter bounds compared to traditional linear relaxations in MILP, a notable improvement in robustness certification accuracy.

3. **Efficient Computation**: Experimental results demonstrate considerable speedups (up to 50x in severe attack scenarios), showing that the method is computationally efficient and less conservative in bounding compared to LP-based techniques, especially in high-dimensional BNNs.

4. **Broad Norm Compatibility**: The method accommodates both ∥.∥∞ and ∥.∥2 norms for adversarial attacks, expanding its applicability across different attack types, which is less common in the BNN verification field.

**Weaknesses:**

1. **Limited Dataset and Network Complexity**: The experiments are primarily on MNIST-based networks, which may not fully demonstrate the method’s performance on more complex datasets or larger architectures. Expanding the experimental validation could strengthen the generalizability of the approach.

4. **Comparative Analysis with State-of-the-Art**: While comparisons with LP and MILP are made, a broader comparison with recent state-of-the-art BNN verification techniques could further validate the advantages and potential limitations of the SDP-based approach.

**Questions:**

1. Have you tested this method on more complex datasets or architectures beyond MNIST? If so, what were the results, and if not, do you anticipate any challenges?

---

> ### Author Response · Authors · 2024-11-23
> **Detailed Response to Reviewer Rj7X**
>
> We are grateful to the reviewer for the nice summary and for acknowledging the strengths of our work. Below, we address all the raised weaknesses and questions in order in which they appear.
>
> ### **Weaknessess:**
>
> - We refer the reviewer to the **Common Reply** section above, where we presented further results showcasing the performance of our method on the CIFAR-10 dataset. As indicated by those results, our method continues to demonstrate notable efficiency compared to MILP in providing *certifiable robustness guarantees*.
>
> - There are several factors that influenced the design of our experimental setup:
>
>   - *First*, our approach is a *relaxation-based* and grounded in optimization, as opposed to the more commonly used SAT/SMT-based methods, which focus on *exact* verification.
>   - *Second*, to the best of our knowledge, there are currently no other scalable methods, specifically tailored for BNN verification that effectively handle continuous input spaces while supporting both $||.||_\infty$ and $||.||_2$ adversarial attacks.
>
>    While the SAT/SMT approach proposed by (Amir *et al.*, 2021) supports continuous input spaces, its performance has only been demonstrated on very small networks with a total of 300 hidden layers.
>    Similarly, MILP-based methods (Lazarus \& Kochenderfer, 2022) depend on a specific encoding of the sign function, which, as we show in Remark 3.1, is less efficient than the MILP formulation used in our benchmarks.
>
> ### **Questions:**
>
> - As we have argued in **Common Reply** section and other answers above, our relaxation-based method yields optimization problems with a polynomial number of constraints in the number of (clique) variables, suggesting better efficiency for larger-scale problems compared to SAT/SMT-based methods. While bound quality may degrade with network depth, Theorem 4.1 and Table 3 (Appendix A.3) indicate that our bounds would still outperform the LP bounds used in MILP approaches. We hope this work inspires further research into scalable SDP solvers or BNN-specific branch-and-bound algorithms to address the limitations of our POP-based approach.

---

### Official Review · Reviewer_TkGW · 2024-11-04

**Soundness:** 3
**Presentation:** 3
**Contribution:** 3
**Rating:** 6
**Confidence:** 3

**Summary:**

The paper presents a novel approach for verifying properties of Binary Neural Networks (BNNs), particularly in the context of robustness against adversarial attacks. Traditional verification methods for BNNs, such as Satisfiability Modulo Theories (SMT) and Mixed-Integer Linear Programming (MILP), face scalability issues when applied to larger networks. To address these challenges, the authors propose using Semidefinite Programming (SDP) relaxations derived from sparse polynomial optimization. This approach is designed to verify BNN robustness efficiently and accurately, overcoming numerical challenges inherent in MILP solvers. Experimental results indicate that the SDP-based method provides significant improvements in both robustness verification against adversarial attacks and computational efficiency, with an average speedup of 4.5 to 11.4 times compared to conventional methods.

**Strengths:**

-  Introduces a new SDP-based approach that enhances the scalability and precsion of BNN verification.
 - Efficiently handles continuous input spaces without requiring input quantization.
 -  Theoretical contributions, including tighter SDP relaxations, improve the accuracy of robustness bounds.
  - Experimental validation across benchmarks highlights the method’s advantages in speed and robustness certification.

**Weaknesses:**

The paper presents an interesting approach; however, it lacks sufficient model variety in its experiments. Only two models were used to demonstrate the proposed method, which limits the generalizability and persuasiveness of the results. For a more robust evaluation, it would be beneficial to include additional models, particularly from diverse architectures, to strengthen the findings and validate the method across a broader range of scenarios.

**Questions:**

Could the authors provide additional experimental results on a wider variety of models? Including more model architectures would enhance the robustness of the conclusions and provide a stronger case for the method’s effectiveness across different settings.

**Details Of Ethics Concerns:**

/

---

> ### Author Response · Authors · 2024-11-23
> **Detailed Response to Reviewer TkGW**
>
> We are grateful to the reviewer for the nice summary and for acknowledging the strengths of our work. Below, we address all the raised weaknesses and questions in order in which they appear.
>
> ### **Weaknesses:**
>
> - One of the primary objectives of our work was to highlight the potential of combining sparse SDP relaxations with redundant constraints (tautologies) to enhance the robustness verification process for binary neural networks (BNNs).  \
> Our focus was primarily on feed-forward neural networks, as they often serve as the foundational building blocks for more complex architectures.
> However, our approach is adaptable and can be extended to more advanced architectures, such as convolutional neural networks (CNNs), since operations like max-pooling and average pooling can be formulated in a semi-algebraic manner. Exploring these directions represents a promising avenue for future research.
>
> ### **Questions:**
>
> - We refer the reviewer to the **Common Reply** section and our response to the previous point.

---

### Author Response · Authors · 2024-11-23
**Common Reply - Part 3**

## __Conclusion:__
 For more detailed responses regarding all specific comments, we refer the reviewers to the individual response sections.

 We greatly appreciate the constructive feedback and will incorporate these insights to enhance the clarity, completeness, and applicability of our work.

---

### Author Response · Authors · 2024-11-23
**Common Reply - Part 2**

### -  __MILP-related:__
Reviewers 7uz5 and o79c  raised concerns about the performance superiority of MILP methods in certain scenarios. MILP/MINP methods are exact (capable of proving both robustness and non-robustness), thus it is not surprising to observe them solving more instances within relatively long  time. Despite this, as displayed in Table 1 and Table 2, the gap in terms of number of solved instances significantly reduces for more severe or  $||.||_2$ attacks, favoring our method.


To further demonstrate the versatility of our method, we provide additional illustrative experiments involving more complex data sets and larger scale optimization problems, as suggested by reviewers TkGW and Rj7X. More precisely, we introduce  $BNN_3: [3072,5000,800,10], w_s=55.97$%, achieving  the test accuracy of $47.66$% on CIFAR-10 data set.


**Table 1: Verifying robustness of BNN_3 on CIFAR-10, for an input region determined by $\delta_{||.||_\infty}=0.2/255$. Data were scaled to $[-1,1]^{3072}$, and a time limitation of $3600 s$ was used.  We present the results of the robustness verification queries on the first $40$ images from the test data set, among which $22$ were correctly classified.**


| Image Index |$\tau_\text{tighter, cs}^{1}$ - bound | $\tau_\text{tighter, cs}^{1}$ - $t\phantom{1}(s)$ | $\tau_{\text{Soft-MILP}}$ - bound | $\tau_{\text{Soft-MILP}}$ - $t\phantom{1}(s)$ |
| --- | --- | --- | --- | --- |
1  | **20.46 (robust)** |  1179.55  |timeout | $>3600$  |
2  | **7.83 (robust)** |   571.08  |timeout | $>3600$  |
7 | -30.28 (unknown) | 380.10 |timeout |$>3600$ |
8 |-68.50 (unknown) | 1721.79 |feasible (not robust) |1.71 |
10 |-62.31 (unknown) | 1330.03 |feasible (not robust) |87.46 |
11 |-142.50 (unknown) |650.79 |feasible (not robust) |0.11 |
12 |-94.92 (unknown) |1115.18 |feasible (not robust) |0.071 |
14 |-103.36 (unknown) |493.80 |feasible (not robust) |0.067 |
15 |-71.09 (unknown) |2090.44 |feasible (not robust) |57.09 |
18 |-97.96 (unknown) |1469.67 |feasible (not robust) |0.08 |
19 |31.06 (robust) | 344.70 |infeasible (robust) |9.02 |
20 |-25.66 (unknown) |823.54 |timeout |$>3600$ |
21 |-90.54 (unknown) |670.94 |feasible (not robust) |0.59 |
24 |-29.08 (unknown) |629.50 |timeout |$>3600$ |
27 |-83.76 (unknown) |766.95 |feasible (not robust) |0.54 |
29 |-110.26 (unknown) |873.51 |feasible (not robust) |0.07 |
30 |-60.63 (unknown) |1407.68 |feasible (not robust) |1.39 |
31 |-29.51 (unknown) |1190.56 |timeout |$>3600$ |
33 |timeout |$>3600$ |timeout |$>3600$ |
34 |-73.15 (unknown) |569.46 |feasible (not robust) |0.51 |
35 | **36.62 (robust)** |$ 657.73 $ |timeout |$>3600$  |
40 |-52.82 (unknown) |650.79 |timeout |$>3600$ |




As presented in the supplementary Table 1, our approach demonstrates comparable performance even on larger datasets, such as CIFAR-10, and for larger networks involving nearly 9000 neurons.

Specifically, $\tau_{tighter, cs}^{1}$ proves the robustness of images 1, 2 and 35 at least 3x, 6x, and 5.5x faster, respectively. In contrast, the low quality of LP bounds prevents MILP from providing an answer within a 1-hour time limit. These additional experimental results confirm that our method keeps providing high-quality lower bounds even, even for large-scale problems, which is consistent with the results presented in Table 3 of our paper.

However, for instances 8 and 10, our method is unable to provide an answer, while MILP can certify **non-robustness** efficiently. This difference arises because MILP methods do not solve the optimization problems to optimality but instead focus on determining the feasibility of an attack, which is inherently less demanding.


We believe that incorporating our tighter SDP bounds within the MILP framework could enhance the ability of MILP methods to certify **robustness** in more complex cases. This represents a promising direction for future research.

We will include a more comprehensive set of large-scale experiments in the subsequent versions of our work.

We would also like to emphasize that we did not require branch-and-bound based MILP/MINP methods to compute *exact* optimal values of the involved optimization problems, as it would have resulted in much more timeouts, making this approach impractical. Rather, we required the MILP/MINP solver to stop as soon as the nonnegative bound could be obtained, while the SDP solver was required to compute $\tau_{tighter,cs}^1$ exactly.

To conclude, our approach should be understood to complementary to the exact MILP/MINP approach, especially for providing robustness certificates (not for finding valid adversarial attacks) in cases where the increased complexity of exact computations prevents MILP/MINP from providing a valid answer. Since availability of precise bounds is essential for robustness verification, we believe that our proposed tight and reliable bounds would yield promising result if efficiently coupled with MILP/MINP methods.

---

### Author Response · Authors · 2024-11-23
**Common Reply - Part 1**

## __Strengths:__

We greatly appreciate the reviewers’ recognition of the key strengths of our work.

Reviewers generally highlighted the relevance of research works attempting to develop scalable verification algorithms that leverage the unique structure of BNNs, avoiding the complexity of exact SAT/SMT/MILP methods.


Both Reviewers o79c and TkGW  noted the strong theoretical foundation and empirical evidence supporting our tight first-order SDP relaxation.


Moreover, reviewers also appreciated the novelty of our method, as we are the first to employ SDP relaxations for BNN verification, effectively exploiting the sparse structure of BNNs and
providing reliable and tighter lower bounds capable of certifying robustness.


The ability to handle $||\cdot||_2$ -attacks and continuous input data was also noted as a significant strength, notably by the reviewers 079c, Rj7X and TkGW.

## __Limitations:__
 We would also like to express our gratitude to all the reviewers for their meaningful critiques and suggestions. Below, we address some common points and concerns that were raised.
### - __Additional experiments:__
As pointed out by reviewers TkGW and Rj7X, we acknowledge the need for results on deeper networks with more layers and more complex data sets to better understand scalability. At the same time, we are not aware of any scalable methods specifically tailored for BNN verification that can successfully operate on continuous input spaces and handle both $||\cdot||_\infty$ and $||\cdot||_2$ attacks. Our relaxation-based method yields optimization problems with number of constraints being polynomial in the number of (clique) variables, which gives us confidence that for larger-scale problems, it would demonstrate significantly better efficiency compared to existing algorithms such as exact SAT/SMT-based methods. Moreover, we anticipate that while the quality of our bound may degrade with increased network depth, it would still outperform the LP bound used in the MILP approach, as supported by Theorem 4.1 and insights from Table 3 in Appendix A.3.

Finally, we have included additional experiments (see **Table 1** in **Common Reply - Part 2**)  from the CIFAR-10 data set, with another network of more than 9000 nodes, which is quite important with respect to other methods relevant to our framework. We observe the similar type of behavior of our method - it can provide robustness guarantees for instances where MILP fails due to the extremely increased complexity.

We hope that our work can inspire further research into developing easily parallelizable and scalable SDP solvers, or BNN-verification-tailored branch-and-bound algorithms, as this would address the raised limitations of our POP-based approach.

### - **Continuous input and $||\cdot||_2$-robustness**
 Relevance of $||\cdot||_2$ attacks and continuous input spaces was questioned by the reviewer 7uz5. However, verifying $||\cdot||_2$ robustness of  neural networks has been widely studied in the standard neural network literature since $||\cdot||_2$ perturbations allow to model attacks with coordinate inter-dependence, which is typical in many real-life cases (blurring, noise, etc...). Additional complexity introduced by these non-linearities prevents most currently available BNN verification tools from being applicable without significant modifications.

Verifying BNNs on continuous input spaces is uniquely challenging due to joint impact of binary weights, discontinuous activation functions, and increased sensitivity to floating-point errors induced by non-discretization. All these factors make the verification problem highly combinatorial and ill-conditioned, as noted on Line 222. Input quantization, while improving scalability, sacrifices precision, often leading to false negatives or inconsistent results depending on the discretization level. In contrast, our method avoids artificial discretization and provides reliable bounds accounting for all possible input states.

---

### Author Response · Authors · 2024-12-04
**Post-rebuttal Summary**

Dear Reviewers,

We would like to use this opportunity to provide a short summary following the discussion period.

First of all, we thank you all for your time, valuable feedback and constructive comments. It has been very useful for enhancing the overall quality of our paper.

Secondly, we express our sincere gratitude to the Reviewers for recognizing the strengths of our contribution.

Specifically, Reviewers 7uz5 and o79c highlighted the significance of addressing the BNN robustness verification problems, while all Reviewers acknowledged the novelty of our approach, which is the *first  SDP relaxation based* BNN verification framework that efficiently exploits the sparse structure of the network. Moreover, our proposed method has a strong theoretical foundation and has been supported experimentally, which has been emphasized by Reviewers  TkGW, Rj7X and o79c.

Furthermore, our method does not rely on input quantization and is capable of handling various types of adversarial attacks, such as $||\cdot||_2$-norm attacks, which have been studied in the context of general DNN verification, but have never been discussed in BNN verification literature. This distinctive feature in the domain of BNN verification was particularly recognized by Reviewers Rj7X, TkGW, and o79c.

We have also addressed the following important points raised by the Reviewers:

- *Additional results:* Reviewers Rj7X, TkGW, and 7uz5 suggested incorporating additional experiments and comparisons to further substantiate our findings. In response, we included Table 8, containing new results about verification queries from the CIFAR-10 dataset. These results confirm our previous conclusions: exact methods, such as MILP, may be efficient for finding adversarial attacks but *cannot provide certified robustness guarantees* within a relatively short time limitation due to their reliance on extremely coarse LP bounds. Similarly, other exact methods, such as the current SAT-based approaches, are highly dependent on heuristic simplifications introduced by input discretization and assumptions of attack linearity.
- *Continuous input:* In *Detailed Response to Reviewer 7uz5 - Continuation 2*, we have provided stronger arguments supporting the importance of verifying BNNs on continuous input spaces. Indeed, BNNs are typically trained and deployed on continuous data, underscoring the need for verification methods that are compatible with continuous input as well. Input quantization, while used as an artificial heuristic to enhance scalability, fails to account for all possible input configurations.
- *Other remarks:* The revised version of our paper contains many improvements in phrasing, notations, and overall clarity, which was recognized by the Reviewer o79c.

We believe that these revisions and additional explanations address your concerns and significantly enhance the clarity and completeness of our paper. Furthermore, we are confident that our work has the potential to convince the research community of the necessity of developing more efficient optimization solvers, thereby facilitating the resolution of many important industrial problems, including the robustness verification of BNNs.

Sincerely,

The Authors

---

### Meta-Review · Area_Chair_aM7h · 2024-12-22

**Metareview:**

The paper proposes a new approach for verifying binary neural networks. It is based on a novel formulation of semidefinite programming (SDP), derived from sparse polynomial optimization. The angle to develop this new formulation is novel, and the paper also includes good insights on how it compares to linear programming and how to strengthen this formulation. The numerical results are also promising, although conducted on limited settings only. The AC believes the theoretical contribution is novel and sufficient for publication.

The AC shares the same opinion as Reviewer o79c that **more related work from the broad neural network verification community should be discussed in this paper** in Section 1.2 (VNN-COMP reports should be good starting points to find relevant references). The current scope of related work is too narrow.

**Additional Comments On Reviewer Discussion:**

Reviewers discussed the paper with the authors, and during the rebuttal period, most technical concerns were addressed. During the discussion between reviewers and the AC, we reached a consensus that this paper could be accepted.

---

### Decision · Program_Chairs · 2025-01-22

Accept (Poster)